# IMLP: An Energy-Efficient Continual Learning Method for Tabular Data Streams

## Abstract

Tabular data streams are rapidly emerging as a dominant modality for real-time decision-making in healthcare, finance, and the Internet of Things (IoT). These applications commonly run on edge and mobile devices, where energy budgets, memory, and compute are strictly limited. Continual learning (CL) addresses such dynamics by training models sequentially on task streams while preserving prior knowledge and consolidating new knowledge. While recent CL work has advanced in mitigating catastrophic forgetting and improving knowledge transfer, the practical requirements of energy and memory efficiency for tabular data streams remain underexplored. In particular, existing CL solutions mostly depend on replay mechanisms whose buffers grow over time and exacerbate resource costs.

We propose a *context-aware incremental Multi-Layer Perceptron (IMLP)*, a compact continual learner for tabular data streams. IMLP incorporates a windowed scaled dot-product attention over a sliding latent feature buffer, enabling constant-size memory and avoiding storing raw data. The attended context is concatenated with current features and processed by shared feed-forward layers, yielding lightweight per-segment updates. We evaluate IMLP against state-of-the-art (SOTA) tabular models on real-world concept drift benchmark tabular datasets designed to assess models under temporal distribution shifts. Compared to TabPFNv2 under the *incremental* concept drift, IMLP has 22.7% total energy reduction while only a 0.05 final balanced accuracy drop. The results show that the proposed attention-based feature memory design can effectively guide the energy consumption while achieving the highest final accuracy in the abrupt concept drifts among all network baselines.

## 1 Introduction

Tabular data, structured as a collection of features and instances, is one of the most common and practical data types in practical machine learning applications, for example, in both high-stakes domains and lower-stakes domains (Amrollahi et al., 2022; Ramjattan et al., 2024; Li et al., 2025b). As such domains increasingly rely on streaming data sources, tabular data streams are gaining significant attention due to their ability to capture continuous, real-time updates rather than static snapshots (Borisov et al., 2022). In particular, most such scenarios often occur on edge devices, IoT systems, and mobile platforms, where energy budgets, battery life, and computational resources are severely constrained Chang et al. (2021).

To tackle those real-world dynamics, Continual Learning (CL) (Wang et al., 2024a), also referred to as lifelong learning (Lee & Lee, 2020), enables models to incrementally acquire, update, accumulate, and exploit knowledge over time. While significant progress has been made on overcoming catastrophic forgetting (Kemker et al., 2018; Li et al., 2019; Bhat et al., 2022) and knowledge transfer (Ke et al., 2021; Li et al., 2024; Shi et al., 2024a), much less is known about their computational analysis and energy efficiency (Li et al., 2023; Trinci et al., 2024).

Energy-efficient continual learning (EECL) has become a practical necessity for real-world applications that require real-time adaptation on resource-constrained platforms (Chavan et al., 2023; Shi et al., 2024b; Trinci et al., 2024; Xiao et al., 2024). Meanwhile, most CL progress to date targets image (Trinci et al., 2024; Chavan et al., 2023; Shi et al., 2024b) and language tasks (Li et al., 2025a; Wang et al., 2024b). In contrast, tabular data streams remain underexplored. Tabular models that

excel on static datasets do not transfer directly to non-stationary streams with tight memory, compute, and energy budgets. Existing CL methods rarely target these constraints. In particular, replay-based strategies rely on buffers that grow over time, increasing storage and compute, and hindering on-device deployment. This gap motivates methods for tabular streaming CL that sustain accuracy under distribution shift while operating at low energy cost, with fixed memory, and without storing raw examples. Moreover, trade-offs between energy consumption and predictive performance matter in lower-stakes domains, especially when the cost of electricity is taken into account. Achieving this under strict resource budgets while mitigating catastrophic forgetting remains a central challenge for Green AI (Henderson et al., 2020; Bouza et al., 2023; Trinci et al., 2024; Różycki et al., 2025).

This paper introduces *Incremental Multi-Layer Perceptron (IMLP)*, a novel method for energy-efficient continual learning, particularly focusing on tabular data streams. IMLP augments a simple MLP with self-attention capabilities, while maintaining efficiency in compute, memory, and energy usage. To be specific: 1) IMLP employs a windowed scaled dot-product attention with a sliding feature buffer, enabling the model to adaptively attend to the most relevant parts of the stream while storing only latent features without needing to revisit raw historical data. 2) The resulting attended representation is concatenated and passed through two shared feed-forward layers followed by a classifier head, serving as the MLP learner for classification tasks. This design avoids the unbounded memory growth inherent to replay baselines (Rebuffi et al., 2017; Li & Hoiem, 2017; Lopez-Paz & Ranzato, 2017), while remaining computationally lightweight on resource-constrained devices. To evaluate hardware-grounded energy-accuracy trade-offs in CL on tabular data streams, we provide quantitative Pareto AUC and global efficiency analysis.

## 2 RELATED WORK

Traditional tabular data models can be roughly categorized into three main groups: Gradient-Boosted Decision Trees (GBDTs) (Friedman, 2001), Neural Networks (NNs) (Goodfellow et al., 2016), and classic models (e.g., SVMs (Cortes & Vapnik, 1995), k-NN (Cover & Hart, 1967), linear model (Cox, 1958), and simple decision trees (Loh, 2011)).

**GBDTs and their variants for CL.** Traditional GBDTs such as XGBoost (Chen & Guestrin, 2016), LightGBM (Ke et al., 2017), and CatBoost (Prokhorenkova et al., 2019) remain strong baselines for tabular classification due to their efficiency and robustness, especially on large or irregular static datasets. However, they are not naturally suited for CL: (1) new data typically requires retraining from scratch, since tree splits and boosting weights depend on the full dataset (Chen & Guestrin, 2016; Ke et al., 2017; Prokhorenkova et al., 2019); (2) without access to past data, models trained only on new samples overwrite previous knowledge, causing catastrophic forgetting (Wang et al., 2024a); and (3) unlike NNs, GBDTs lack mechanisms for knowledge transfer across tasks (Ke et al., 2021; Parisi et al., 2019; De Lange et al., 2021). Extensions such as online bagging and boosting (Oza & Russell, 2001) or warm-starting (Pedregosa et al., 2011), and adaptive XGBoost (Montiel et al., 2020), partially mitigate these issues, but remain limited in long-term knowledge retention due to the lack of representation reuse, especially when compared to neural CL methods.

**Classic models in CL.** Both standard SVMs (Cortes & Vapnik, 1995) and decision trees (Loh, 2011) are batch learners, requiring retraining on the full dataset when new tasks arrive. SVMs can be extended to CL through incremental or online variants such as incremental SVM (Cauwenberghs & Poggio, 2000), LASVM (Bordes et al., 2005), and NORMA (Kivinen et al., 2004), which handle streaming updates but still face challenges with scalability, memory growth, and forgetting. k-NNs (Cover & Hart, 1967) trivially avoid forgetting if all data is stored, but this violates the constraint of no access to past raw inputs and is impractical under resource limits. Linear models (Cox, 1958) are efficient but prone to forgetting under distribution shifts, as updates overwrite prior knowledge. Incremental decision trees, such as Hoeffding Trees (Domingos & Hulten, 2000), and streaming ensembles (Bifet et al., 2010; Gomes et al., 2017) can adapt to data streams without full retraining. Still, their accuracy degrades under severe drift, since they lack strong representation learning, and ensemble methods can be computationally expensive.

**Neural models in CL.** Recent studies demonstrate that advanced NNs (Zabërgja et al., 2024; Arik & Pfister, 2021; Kadra et al., 2021; Gorishniy et al., 2023a; Hollmann et al., 2025b; Ye et al., 2024; Gorishniy et al., 2024) can surpass GBDTs on static tabular data in certain regimes, e.g., with well-regularized MLPs (Kadra et al., 2021), attention-based models such as SAINT (Somepalli et al., 2021),

or meta-learned foundation models like TabPFN and its variants (Hollmann et al., 2025b). While their training is typically computationally intensive than that of GBDTs unless carefully tuned (Kadra et al., 2021), NNs are generally better suited for streaming data, owing to their rich representations, incremental updates via stochastic gradient descent, and flexible architectures. However, vanilla NNs still suffer from catastrophic forgetting in the absence of CL strategies (Wang et al., 2024a).

**CL strategies with neural models.** In NNs, CL strategies are commonly categorized into regularization-based approaches (Kirkpatrick et al., 2017; Zenke et al., 2017), replay-based strategies (Rebuffi et al., 2017; Shin et al., 2017), attention-based retrieval mechanisms (Chaudhry et al., 2019; Aljundi et al., 2017), and architectural methods (Rusu et al., 2016). Regularization-based methods, such as EWC (Kirkpatrick et al., 2017), SI (Zenke et al., 2017), MAS (Aljundi et al., 2017), and LwF (Li & Hoiem, 2016), mitigate forgetting by constraining updates to parameters deemed important for previously learned tasks. Replay-based strategies, including iCaRL (Rebuffi et al., 2017) and generative replay (Shin et al., 2017), maintain past knowledge by rehearsing stored samples or synthetic data. Attention-based retrieval mechanisms, such as A-GEM with attention (Chaudhry et al., 2019) and attentive experience replay (Aljundi et al., 2017), employ attention to prioritize and retrieve relevant past experiences. Architectural methods, exemplified by PNNs (Rusu et al., 2016), expand model capacity by freezing previously trained components and introducing new modules for incoming tasks.

Despite recent progress, energy-efficient CL for tabular data streams remains largely unexplored (Chavan et al., 2023; Trinci et al., 2024). Real-world tables frequently undergo domain drift (e.g., quarterly finance transactions, evolving sensor logs, healthcare data) without changes to the label space. Yet, no standardized domain-incremental learning benchmark that considers energy-performance trade-offs currently exists for tabular streams. Moreover, pre-trained transformers for tabular data (Gorishniy et al., 2023b; Hollmann et al., 2025b) and feature-level or attention-based CL strategies (Pellegrini et al., 2020; Vaswani et al., 2017a; Jha et al., 2023) show promise for low-storage, privacy-preserving CL, but their effectiveness under domain drift has not been systematically evaluated. Here, we bridge this gap by introducing our method, establishing fair comparisons, and quantifying energy–performance trade-offs.

# 3 IMLP: AN INCREMENTAL MLP FOR TABULAR DATA STREAMS

Owing to the general difficulty and diversity of challenges in CL, we focus on a simplified task incremental learning setting (Parisi et al., 2019; De Lange et al., 2021). In this setting, a model is trained on a sequence of tasks $\{\mathcal{T}_t\}_{t=1}^T$, where the data for each task arrives incrementally at time $t$.

## 3.1 PROBLEM STATEMENT

**Problem Setup.** We consider a sequence of tasks $\{\mathcal{T}_t\}_{t=0}^T$, where the training data for each task arrives incrementally at time $t$. Each task $\mathcal{T}_t$ is associated with data $(\mathcal{X}_t, \mathcal{Y}_t)$ randomly drawn from distribution $\mathcal{D}_t$, where $\mathcal{X}_t$ denotes the set of data samples and $\mathcal{Y}_t$ is the corresponding ground truth labels. Our goal is to design an incremental learner $f_\theta$ that updates online and minimizes the expected risk $\hat{L}_t(\theta)$ across all observed tasks, with limited or no access to the data from earlier tasks $t < T$,

$$\hat{L}_t(\theta) := \sum_{t=0}^{T} \mathbb{E}_{(\mathcal{X}_t, \mathcal{Y}_t) \sim \mathcal{D}_t}[\ell_t(\theta)], \tag{1}$$

where $\ell_t(\theta)$ represents the loss function of the model $f_\theta(\mathcal{X}_t, H_t)$ with input $\mathcal{X}_t$, parameter $\theta$, and the historical features $H_t$ at time $t$. Additionally, we aim to achieve energy efficiency.

**Assumptions.** We formalize this with standard non-convex optimization assumptions for NNs.

**(A1)** *There exists $R_\mathcal{X} > 0$ such that $\|\mathcal{X}_t\|_2 \leq R_\mathcal{X}$ for all samples in the arrived stream $\mathcal{T}_t$.*

**(A2)** *The precomputed latent features are $\ell_2$-normalized, i.e., $\|h_{t,j}\|_2 \leq 1$ for all $t, j$.*

**(A3)** *Training is performed with weight decay and early stopping, so that for some $R_\theta > 0$, $\|\theta\|_2 \leq R_\theta$ throughout optimization.*

## 3.2 ARCHITECTURE OVERVIEW

For efficient learning from the current task $\mathcal{T}_t$ while maintaining performance on previously learned tasks, we propose an incremental multi-layer perceptron (IMLP) architecture, as shown in Figure 1. We employ two strategies: (1) processing each task with an augmented MLP learner module $\mathcal{M}^{1_{\text{att}}}$

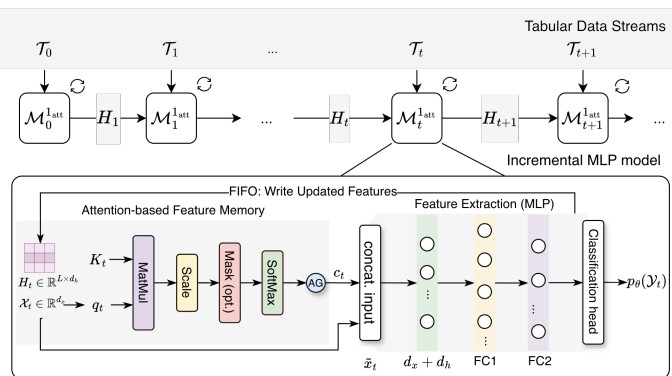

Figure 1: IMLP architecture. IMLP sequentially takes $\mathcal{T}_t$ as raw input and outputs predictive performance $p_\theta(\mathcal{Y}_t)$.

that incorporates limited historical context $H_t$ in a window size $W$ through a variant of scaled dot-product attention (Vaswani et al., 2017b); and (2) maintaining an FIFO feature buffer with fixed memory over time to handle the concept drifts (Hoens et al., 2012), which facilitates representation reuse while keeping memory and computation cost constrained as new data evolves.

Given the current input $\mathcal{X}_t \in \mathbb{R}^{d_x}$, a hidden dimension $d_h$, and a FIFO memory of the past $L$ features. We denote by $H_t = [h_{t,1}, \ldots, h_{t,L}]^\top \in \mathbb{R}^{L \times d_h}$ the matrix that stacks the latent features in the window associated with time $t$, with learnable maps $W_q \in \mathbb{R}^{d_h \times d_x}$ and $W_k \in \mathbb{R}^{d_h \times d_h}$, IMLP forms

$$q_t = W_q \mathcal{X}_t + b_q \in \mathbb{R}^{d_h}, \tag{2}$$

$$K_t = H_t W_k^\top + \mathbf{1}_L b_k^\top \in \mathbb{R}^{L \times d_h}, \tag{3}$$

$$s_t = K_t q_t \in \mathbb{R}^L, \tag{4}$$

$$\tilde{s}_t = \frac{1}{\sqrt{d_h}} s_t, \tag{5}$$

$$\alpha_t = \text{softmax}(\tilde{s}_t) \in \mathbb{R}^L, \tag{6}$$

where

$$\alpha_{t,j} = \frac{\exp(\tilde{s}_{t,j})}{\sum_{j=1}^L \exp(\tilde{s}_{t,j})} \quad \text{for } j = 1, \ldots, L (L \le W), \tag{7}$$

and $\mathbf{1}_L \in \mathbb{R}^L$ is the all-ones vector used to broadcast the bias $b_k$ to all $L$ rows. $h_{t,j} \in \mathbb{R}^{d_h}$ is the feature vector of the $j$-th most recent sample before $\mathcal{X}_t$ (with $j = 1$ being the most recent). The $j$-th row $K_{t,j} \in \mathbb{R}^{d_h}$ is the key for the $j$-th past feature in the window.

The attention-based feature memory is the weighted sum of the keys $c_t = \alpha_t^\top K_t$. Finally, IMLP concatenates the context with the current input $\tilde{x}_t = [\mathcal{X}_t, c_t]^\top \in \mathbb{R}^{d_x + d_h}$ and then feeds it to the feature extractor $f_{t,\theta} = \phi(\tilde{x}_t) \in \mathbb{R}^{d_h}$, where $\phi(\cdot)$ is a two-layer MLP, and the classifier outputs $\hat{y}_t = W_{cls} \cdot f_{t,\theta} + b_{cls}$ where $W_{cls} \in \mathbb{R}^{C \times d_h}$ and $b_{cls}$ denotes the weight matrix and bias, respectively. This corresponds to the model's performance $p_\theta(\mathcal{Y}_t)$ at the time $t$.

In the following, we detail the properties of the proposed attention-based feature memory design to achieve EECL over tabular data streams.

## 3.3 CONVERGENCE ANALYSIS OF IMLP

Let $\mathcal{A}(\mathcal{X}_t, H_t; \theta_{\text{att}}) := c_t$ denote the attention-based context, where $\theta_{\text{att}} = (W_q, b_q, W_k, b_k)$ and $H_t$ collects the latent features in the window at the time $t$.

**Lemma 3.1** (Bounded context vector). *Under* (A2)*, there exists $B_c > 0$ (depending only on $W_k$ and $b_k$) such that $\|c_t\|_2 \leq B_c$ for all $i$.*

**Lemma 3.2** (Smooth attention map). *Under* (A1)–(A3)*, the map $(\mathcal{X}_t, H_t, \theta_{att}) \mapsto \mathcal{A}(\mathcal{X}_t, H_t; \theta_{att})$ is continuously differentiable, and its Jacobian with respect to $\theta_{att}$ is bounded on the compact set*

$$\mathcal{K} := \{(\mathcal{X}_t, H_t, \theta_{att}) : \|\mathcal{X}_t\|_2 \leq R_{\mathcal{X}}, \ \|h_{t,j}\|_2 \leq 1, \ \|\theta_{att}\|_2 \leq R_\theta\}. \tag{8}$$

*In particular, there exists $L_{att} > 0$ such that*

$$\|\mathcal{A}(\mathcal{X}_t, H_t; \theta_{att}^{(1)}) - \mathcal{A}(\mathcal{X}_t, H_t; \theta_{att}^{(2)})\|_2 \leq L_{att}\|\theta_{att}^{(1)} - \theta_{att}^{(2)}\|_2, \tag{9}$$

*for all $\theta_{att}^{(1)}, \theta_{att}^{(2)} \in \mathcal{K}$.*

Correspondingly, the full network can be written as

$$f_\theta(\mathcal{X}_t, H_t) := W_{cls} \phi_\theta\big([\mathcal{X}_t; \mathcal{A}(\mathcal{X}_t, H_t; \theta_{att})]\big) + b_{cls}, \tag{10}$$

where $\phi_\theta$ is the two-layer ReLU feature extractor, and the per-sample loss is

$$\ell_t(\theta) = \text{CE}\big(\text{softmax}(f_\theta(\mathcal{X}_t, H_t)), \mathcal{Y}_t\big). \tag{11}$$

**Lemma 3.3** (Smooth network and loss). *Under* (A1)–(A3)*, $f_\theta(\mathcal{X}_t, H_t)$ is continuously differentiable with bounded Jacobian on $\{\theta : \|\theta\|_2 \leq R_\theta\}$, and $\ell_t(\theta)$ has Lipschitz-continuous gradient on the same set.*

For a fixed segment (task) $\mathcal{T}_t$ with data $(\mathcal{X}_t, H_t, \mathcal{Y}_t)$, where $\{(\mathcal{X}_{t,i}, H_{t,i}, \mathcal{Y}_{t,i})\}_{i=1}^{n_t}$ denotes the samples in this segment, we define $\hat{L}_t(\theta) := \frac{1}{n_t}\sum_{i=1}^{n_t} \ell_{t,i}(\theta)$.

**Theorem 3.4** (Segment-wise smooth empirical loss). *Under* (A1)–(A3)*, the empirical loss $\hat{L}_t(\theta)$ is (i) bounded below; (ii) continuously differentiable on $\{\theta : \|\theta\|_2 \leq R_\theta\}$; and (iii) has Lipschitz-continuous gradient on this compact set.*

**Corollary 3.5** (Per-segment convergence of IMLP). *Consider optimizing $\hat{L}_t(\theta)$ with a stochastic first-order method (e.g., SGD or Adam) under standard step-size conditions and weight decay, as in our training loop. Then the iterates on segment $\mathcal{T}_t$ converge to a first-order stationary point in the sense that*

$$\|\nabla\hat{L}_t(\theta_k)\|_2 \to 0 \quad as \ k \to \infty, \tag{12}$$

*or, in the practical finite-epoch setting, reach a parameter $\theta^\star$ with small gradient norm $\varepsilon$. In particular, the attention-based feature memory acts as a bounded, smooth transformation of a finite latent window, so IMLP behaves like a standard MLP with an augmented input $\tilde{s}_t$ and inherits the usual segment-wise convergence guarantees of non-convex deep networks.*

> **Remark on non-stationary streams.** The analysis above is *segment-wise*. Under standard assumptions, the attention-based feature memory yields a bounded, smooth network with Lipschitz-continuous gradients, so first-order optimizers converge to a stationary point of the empirical loss on each fixed segment $\mathcal{T}_t$. However, we do not claim convergence to any global limit when the data-generating process is non-stationary across $t$. Instead, the theory guarantees that, conditional on the data observed in each segment, the optimization problem remains well-behaved despite using attention over a finite feature memory.

Proofs of Lemmas 3.1–3.3 and Theorem 3.4 are given in the appendix A.

### 3.4 FIFO ATTENTION-BASED FEATURE MEMORY AND TIME COMPLEXITY

Unlike replay buffers that grow with the number of seen samples, our FIFO memory has constant memory complexity in time. The attention module adds the query layer $W_q \in \mathbb{R}^{d_h \times d_x}, b_q \in \mathbb{R}^{d_h}$ and key layer $W_k \in \mathbb{R}^{d_h \times d_h}, b_k \in \mathbb{R}^{d_h}$, hence, the parameter memory for attention denotes $\mathcal{O}(d_x \cdot d_h + d_h^2)$, which is constant with respect to the stream length and number of tasks.

At time $t$, the FIFO buffer stores $W$ latent feature vectors $H_t \in \mathbb{R}^{W \times d_h}$, it costs $\mathcal{O}(Wd_h)$ memory per stream, independent of how long the stream has run. In the batched implementation, each FIFO buffer holds $W$ feature tensors, each of shape $[B, d_h]$; hence, the runtime memory overhead is $\mathcal{O}(BWd_h)$.

For a single forward step with batch size $B$, the total computational efficiency of IMLP is given by query computation $\mathcal{O}(Bd_xd_h)$, key computation $\mathcal{O}(BWd_h^2)$, attention scores and weights $\mathcal{O}(Bd_xd_h + BWd_h^2 + BWd_h)$, as well as the rest of the network $\mathcal{O}(B(d_x + d_h) \cdot 512)$, where 512 is the feature dimension of the FC1, FC2 layer in the feature extraction module using MLP. Therefore, the incremental cost scales linearly in $B$ and $W$, its per-step computational cost scales as

$$\mathcal{O}(Bd_h(d_x + Wd_h)) = \underbrace{\mathcal{O}(Bd_h(d_x + Wd_h))}_{\text{query and key}} + \underbrace{\mathcal{O}(Bd_h(d_x + Wd_h + W))}_{\text{attention scores and weights}} + \underbrace{\mathcal{O}(B(d_x + d_h)512)}_{\text{feature MLP}}, \quad (13)$$

where for a fixed $B$ and $d_h$, the incremental overhead of IMLP over a vanilla MLP is controlled and linear in the window size $W$.

Therefore, FIFO attention-based feature memory adds $\mathcal{O}(d_xd_h + d_h^2)$ parameters and $\mathcal{O}(BWd_h)$ runtime memory, while its per-step computational cost scales as $\mathcal{O}(Bd_h(d_x + Wd_h))$, yielding constant memory in time with respect to the length of the data stream.

## 3.5 ENERGY EFFICIENCY ANALYSIS OF IMLP

**Energy model and assumptions.** We assume that for a fixed device and implementation, energy is approximately linear in the number of floating-point operations (FLOPs), up to device-specific constants and small overhead. Let $F_{\text{train}}$ denote the number of FLOPs required to perform one forward-and-backward pass of IMLP on a single sample. We adopt a standard abstract energy model with the following assumptions:

**(A4)** *On a fixed hardware platform (GPU/CPU), there exist constants $0 < \eta_{\min} \leq \eta_{\max}$ such that the energy consumed per FLOP lies in $[\eta_{\min}, \eta_{\max}]$.*

**(A5)** T*he additional system overhead per training step (e.g., kernel launches, bookkeeping) is bounded by a constant $E_0$ independent of the sample index and segment.*

**Lemma 3.6** (FLOP complexity per sample). *Let $C$ be the number of classes and $d_{\text{in}}$ be the input dimension. For a single sample $(\mathcal{X}_{t,i}, H_{t,i}, \mathcal{Y}_{t,i})$, the FLOP count of a forward-and-backward step of IMLP satisfies*

$$F_{train} \leq K_{arch}(d_{\text{in}}d_h + Wd_h^2 + d_h^2 + d_hC), \quad (14)$$

*for an architecture-dependent constant $K_{arch} > 0$ that does not depend on $n_t$ or $t$.*

**Theorem 3.7** (Per-segment energy complexity bound). *Consider a segment $\mathcal{T}_t$ with $n_t$ training samples. Under (A4)-(A5) and Lemma 3.6, the total training energy consumed by IMLP on this segment satisfies*

$$E_t^{train} \leq C_{train} E_{\max} n_t(d_{\text{in}}d_h + Wd_h^2 + d_h^2 + d_hC) + C_0, \quad (15)$$

*for some hardware- and implementation-dependent constants $C_{train} > 0$ and $C_0 \geq 0$. Similarly, the inference energy on the test set of size $n^{test}$ admits*

$$E_t^{infer} \leq C_{infer} n^{test}(d_{\text{in}}d_h + Wd_h^2 + d_h^2 + d_hC) + C_0', \quad (16)$$

*with another constant $C_{infer} > 0$ and overhead $C_0' \geq 0$.*

Proofs of Lemma 3.6 and Theorem 3.7 are given in the appendix B.

**Corollary 3.8** (Energy complexity over the full non-stationary stream). *Let the data stream be partitioned into $T$ segments $\{\mathcal{T}_t\}_{t=0}^{T}$ with sizes $\{n_t\}_{t=1}^{T}$. Under the same assumptions as Theorem 3.7, the total training energy over the entire stream satisfies*

$$E_{total}^{train} = \sum_{t=1}^{T} E_t^{train} \leq C_{train} E_{\max} \Big(\sum_{t=1}^{T} n_t\Big)(d_{\text{in}}d_h + Wd_h^2 + d_h^2 + d_hC) + TC_0, \quad (17)$$

*and the total inference energy satisfies*

$$E_{total}^{infer} \leq C_{infer}\, n_{total}^{test}(d_{\text{in}}d_h + Wd_h^2 + d_h^2 + d_hC) + TC_0'. \tag{18}$$

*Theoretically, for a fixed $d_h$, $W$, and $E_{\max}$, both training and inference energy grow at most linearly in the total number of processed examples $\sum_t n_t$ and in the effective model size. The attention-based feature memory only adds the bounded term $Wd_h^2$ and does not change this linear energy scaling.*

> **Remark on the energy complexity bounds.** The bounds above explain two aspects of our empirical observations: (1) on a fixed device, IMLP has a predictable energy profile, scaling linearly with the number of samples and epochs; and (2) The attention-based feature memory contributes a controlled overhead proportional to $Wd_h^2$, which remains small in our IMLP because $W$ and $d_h$ are fixed. Our measured Joule values are therefore consistent with an energy complexity that is linear in the stream size, and the theoretical bounds clarify that this behavior is not specific to a particular dataset, but a structural property of the IMLP architecture and training procedure.

## 4 ENERGY-ACCURACY TRADE-OFFS

In many optimization problems, objectives are inherently conflicting; for instance, improving the accuracy of a NN increases energy consumption or latency. A classical way to study such trade-offs is through Pareto front analysis (Giagkiozis & Fleming, 2014).

Our convergence and energy bounds naturally lead to a bi-objective viewpoint, where we jointly consider predictive performance and energy consumption. For a fixed segment $\mathcal{T}_t$, an IMLP configuration is determined by its architecture $(d_h, W)$ and optimization budget (e.g., number of iterations, learning-rate schedule). Each such configuration yields a pair $\big(E_t(\theta), P_t(\theta)\big)$, where $E_t(\theta)$ denotes the total energy consumed on $\mathcal{T}_t$ and $P_t(\theta) = p_\theta(\mathcal{Y}_t)$ denotes the resulting segment-wise model performance (e.g., balanced accuracy). We say that a configuration $\theta^{(1)}$ *Pareto-dominates* $\theta^{(2)}$ if

$$E_t(\theta^{(1)}) \leq E_t(\theta^{(2)}), \quad P_t(\theta^{(1)}) \geq P_t(\theta^{(2)}), \tag{19}$$

with at least one strict inequality. The *Pareto set*

$$\mathcal{P}_t := \{\theta : \nexists\, \theta' \text{ s.t. } E_t(\theta') \leq E_t(\theta),\ P_t(\theta') \geq P_t(\theta) \text{ and one inequality is strict}\} \tag{20}$$

collects all Pareto-efficient configurations, and its image in the energy–accuracy plane forms the *Pareto frontier*.

Intuitively, on each segment $\mathcal{T}_t$, our smoothness and Lipschitz-gradient assumptions imply that stochastic first-order methods require on the order of $1/\varepsilon^2$ iterations to reach an $\varepsilon$-stationary point of the empirical loss $\hat{L}_t(\theta)$, i.e., $\|\nabla\hat{L}_t(\theta_k)\|_2 \leq \varepsilon$. Each iteration has a bounded computational cost proportional to $d_{\text{in}}d_h + Wd_h^2 + d_h^2 + d_hC$ (Lemma 3.6), and our energy model (Theorem 3.7) shows that energy is proportional to this cost up to device-dependent constants. Combining these results yields the scaling

$$E_t^{\text{train}}(\varepsilon) = \mathcal{O}\left(\frac{d_{\text{in}}d_h + Wd_h^2 + d_h^2 + d_hC}{\varepsilon^2}\right). \tag{21}$$

Thus, reducing the optimization tolerance $\varepsilon$ leads to a more than linear increase in training energy, with the rate governed by the architectural parameters $(d_h, W)$. For any fixed architecture, attainable pairs $\big(E_t(\theta), P_t(\theta)\big)$ therefore lie on or above a decreasing curve in the energy–accuracy plane: improving performance inevitably requires a disproportionately large increase in energy.

## 5 EXPERIMENTS

**Setup and Configuration.** All experiments were conducted on a single workstation equipped with an Intel® Core™ i5-8600K Processor, a NVIDIA GeForce RTX 2080 Ti GPU, 16GB DDR4 RAM, and

an NVMe SSD for data and model checkpoints. To obtain ground-truth measurements, we instrument our CL pipeline with an ElmorLabs PMD-USB power meter (ElmorLabs, 2023) and corresponding PCI-E slot adapter (ElmorLabs, 2025) for real-life energy consumption measurement.

**Datasets and Baselines.** We evaluate our method on real-world data streams using the River's INSECTS datasets[1], which are specifically chosen to represent challenging concept drift scenarios. The datasets include tasks that exhibit both *abrupt* and *incremental* concept drift as the underlying data distribution changes over time (Souza et al., 2020). We compare our IMLP model against a comprehensive set of seven recent SOTA methods for tabular classification, covering three distinct model categories: 1) foundation models: *TabPFNv2*; 2) deep NN baselines: *RealMLP*, *ModernNCA*, and *MLP*; and 3) GBDTs: *CatBoost*, *XGBoost*, and *LightGBM*. More details are in Appendix C.2.

**Evaluation Protocol.** A crucial consideration for this study is that our selected baselines were primarily developed for static, independent and identically distributed data. While an ideal comparison in our incremental environment would involve creating a dedicated CL variant of each GBDT and NN baseline, e.g., equipped with specialized components for memory and catastrophic forgetting mitigation, such an undertaking is outside the scope of this work. To establish a methodologically sound comparison, we standardize the data flow for all models by applying our FIFO buffer mechanism. This enforces a *segmental training mode* with a limited memory window, thereby comparing IMLP against the SOTA under the same challenging, resource-constrained sequential protocol.

**Statistical Analysis and Metrics.** For a fair evaluation, all datasets undergo the same preprocessing pipeline, with an 85%-15% stratified split used for training and validation/testing within the segmental mode. To assess the statistical significance of performance differences across the evaluated scenarios, we first conduct the Friedman test (Friedman, 1937). If the null hypothesis is rejected, we perform post-hoc analyses using the Wilcoxon signed-rank test (Wilcoxon, 1945) with Holm correction (Holm, 1979), along with critical difference analysis (Nemenyi, 1963). All models are evaluated based on six key metrics: balanced accuracy, log-loss, energy consumption, execution time, and the composite Pareto (AUC and global efficiency) metrics that capture the crucial energy-accuracy trade-offs.

## 5.1 Ablation Study: Impact of Attention, $d_h$, $W$, and Buffer Strategy Choices

**Attention and Buffer Strategy Choices.** We first ablated the core components, i.e., the attention module and buffer strategy, on the *Insects-abrupt-drift* dataset to evaluate the impact of the attention and buffer strategy choices on model performance and energy consumption. We compared our default FIFO strategy against a *similarity-based* strategy, which replaces the most similar feature in the buffer to maximize diversity. The results are presented in Figure 2.

Figure 2a compares models with and without attention, while Figure 2b reports results for different buffer replacement strategies. Enabling attention substantially improves the predictive performance (the median balanced accuracy increases from 0.376 to 0.568) but also raises total energy consumption by approximately 47.11%. The FIFO strategy outperforms the diversity-maximizing similarity strategy, which highlights the

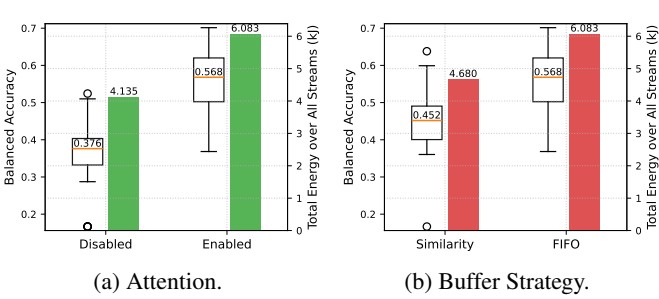

(a) Attention.  (b) Buffer Strategy.

Figure 2: Ablation study. (a) Attention. (b) Buffer strategy.

importance of recency in drifting streams. The results indicate that for handling concept drift in data streams, temporal locality (preserving the most recent samples) is more critical than feature diversity. Meanwhile, the similarity-based buffer strategy has a smaller influence on the total energy consumption compared to the FIFO strategy. Overall, these ablations indicate that the buffer strategy tends to trade additional energy for improved accuracy, while attention acts as the primary lever to control the energy footprint at a given performance level.

---

[1] https://riverml.xyz/dev/api/datasets/Insects/

**Impact of $d_h$ and $W$.** We also ablated the window size $W$ and hidden dimensions $d_h$ on the model performance and total energy cost under the non-stationary distributions, as depicted in Figure 3.

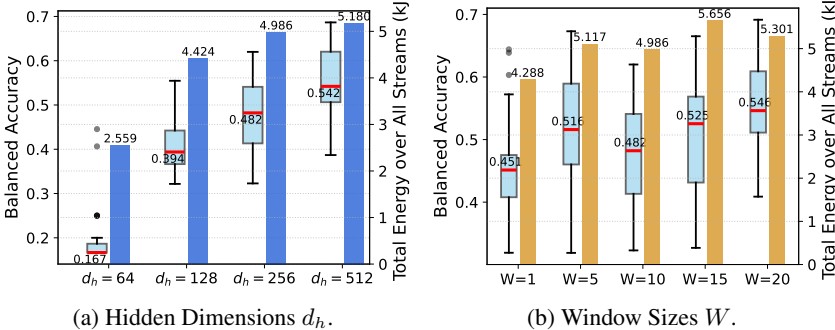

(a) Hidden Dimensions $d_h$.

(b) Window Sizes $W$.

Figure 3: Overview of the impact of $d_h$ and $W$ under the *Insects-abrupt-drift* dataset.

Figure 3a compares with different hidden dimension values ($d_h = 64, 128, 256, 512$) with a fixed window size ($W = 10$), in which the median balanced accuracy improves significantly as the $d_h$ increases from 64 to 512. Correspondingly, the total energy consumption increases from 64 to 512. Figure 3b presents the results for different window sizes ($W = 1, 5, 10, 15, 20$) with a fixed $d_h = 256$. The median balanced accuracy slightly improves as the window size increases, plateauing around $W = 10$ to $W = 20$. However, its upward trend is not linearly dependent on the window size.

Therefore, the *attention-based FIFO feature buffer* module, including the hidden dimension setting, significantly impacts the predictive performance improvement and energy consumption reduction under the non-stationary distributions.

## 5.2 EVALUATION UNDER ABRUPT AND INCREMENTAL DRIFTS

Figure 4 compares the dynamic performance and energy consumed when data arrives in sequence on the *abrupt-* and *incremental-balanced* drift scenarios.

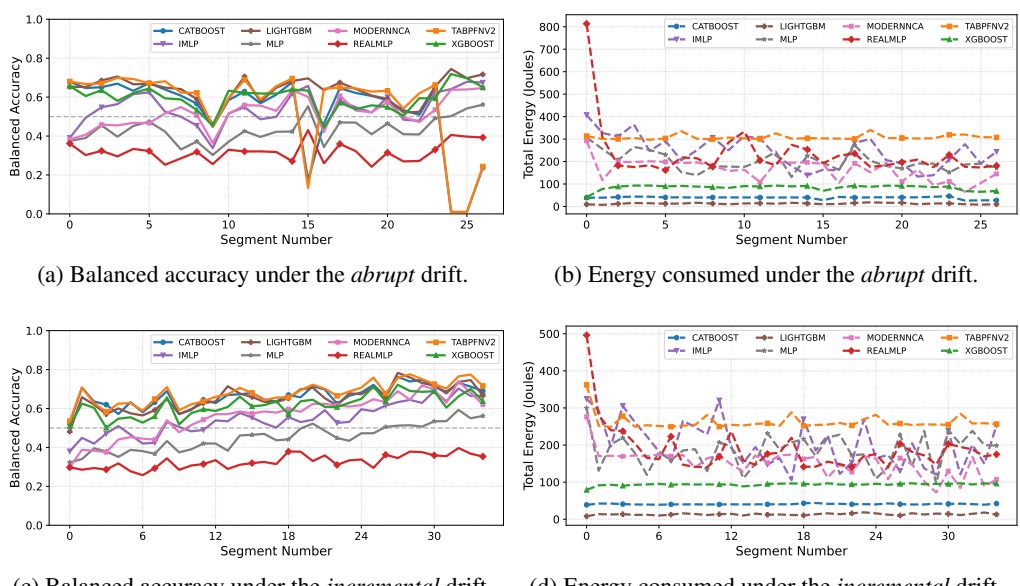

(a) Balanced accuracy under the *abrupt* drift.

(b) Energy consumed under the *abrupt* drift.

(c) Balanced accuracy under the *incremental* drift.

(d) Energy consumed under the *incremental* drift.

Figure 4: Overview of the model performance and energy consumption under different concept drifts.

**Abrupt drifts.** Figure 4a shows that IMLP reacts strongly to abrupt drifts with immediate accuracy degradation, while it also demonstrates a robust ability to recover in the following segment using

the new training data. For example, segments 9 and 16 show significant drops in balanced accuracy (0.338 and 0.436); however, the accuracy recovers to 0.548 at segment 11 and 0.582 at segment 17, respectively, and the final accuracy ranked second (lower than LightGBM equipped with FIFO buffer) among the methods evaluated. The training energy required for such adaptation and recovery tends to be significantly higher than that of the average non-drift segment, as IMLP is effectively retrained from the previous segment's state, which necessitates gradient updates across all layers. Additionally, it must utilize the attention mechanism to analyze features from the memory buffer. Notably, the results show that the FIFO buffer keeps the LightGBM stable and prevents forgetting, while it ensures that this stability is maintained with the lowest computational cost and fastest adaptation speed among the compared models, as shown in Figure 4b.

**Incremental drifts.** Figure 4c depicts that TabPFNv2 (0.716) outperforms under the incremental concept drifts, followed by CatBoost (0.691), LightGBM (0.666), and IMLP (0.666), while it consumes the highest total energy as shown in Figure 4d. Still, GBDTs keep the lowest energy consumption. The IMLP's energy profile is volatile because it is a gradient-based model operating with an aggressive adaptation policy controlled by a performance-based early stopping mechanism. This design means its energy consumption becomes a direct, fluctuating measure of the difficulty of adapting to the new segment's concept. Consequently, most NN baselines (MLP, RealMLP, ModernNCA), when equipped with the FIFO buffer, exhibit similar energy volatility.

**Energy-accuracy trade-offs.** Table 1 presents the trade-off analysis based on final balanced accuracy, total energy consumed, quantitative Pareto AUC, and global Pareto efficiency.

In *abrupt* concept drift, IMLP achieves the highest final balanced accuracy (0.675), while costing 49.7% energy more than that of ModernNCA. Compared to TabPFNv2 under the *incremental* concept drift, IMLP has 22.7% total energy reduction while only a 0.05 final balanced accuracy drop. Both IMLP and ModernNCA remain a global Pareto efficiency of 1.0 in both concept drifts, indicating that they are most often on the neural-global Pareto frontier.

Table 1: Trade-off analysis.

| Data | Method | FinalAcc ($\uparrow$) | TotalEnergy ($\downarrow$) | AUC ($\uparrow$) | Efficiency ($\uparrow$) |
|---|---|---|---|---|---|
| abrupt | TabPFNV2 | 0.244 | 8316.162 | 0 | 0 |
| | RealMLP | 0.393 | 6325.562 | 0.177 | 0 |
| | MLP | 0.562 | _5408.748_ | _0.551_ | 0 |
| | ModernNCA | _0.647_ | **4424.620** | **0.935** | **1.0** |
| | IMLP | **0.675** | 6622.577 | 0.435 | **1.0** |
| incremental | TabPFNV2 | **0.716** | 9159.547 | 0 | **1.0** |
| | RealMLP | 0.354 | 6568.508 | 0 | 0 |
| | MLP | 0.562 | _6500.125_ | 0.396 | 0 |
| | ModernNCA | 0.620 | **5291.275** | **0.737** | **1.0** |
| | IMLP | _0.666_ | 7082.398 | _0.463_ | **1.0** |

**Summary of IMLP's strengths.** IMLP offers several notable advantages over related tabular methods: (1) it is simple and inherently suitable for streaming tabular learning without replaying past raw inputs; and (2) it is lightweight and tunable in both computation and memory, with costs independent of the length of the data stream, yielding an energy-efficient solution.

# 6 CONCLUSION

This paper addresses the critical gap of EECL on tabular data streams by introducing IMLP, a novel incremental MLP model. IMLP employs a novel attention-based feature replay with context retrieval and sliding buffer updates, integrated into a minibatch training loop for streaming tabular learning. Experiments show that IMLP matches the accuracy of neural baselines under no replay while substantially reducing runtime and energy costs. IMLP achieves up to 22.7% energy reduction compared to TabPFNv2, while maintaining competitive average accuracy. Positioned optimally on the neural Pareto frontier, IMLP consistently delivers efficiency gains across abrupt and incremental concept drift datasets.

**Limitations and Future Work.** Despite these exciting findings, IMLP currently treats baselines on River's Insets benchmarks in an experimental setting. A promising next step is to compare the method with up-to-date models on real-life lifelong settings, thereby enriching the benchmarks. Beyond that, building a comprehensive evaluation framework would shed light on the influence of alternative CL strategies for SOTA baselines. Ultimately, an important future direction for EECL is to extend IMLP toward jointly optimizing the trade-offs between energy efficiency and predictive performance with tunable parameters, ideally supported by theoretical guarantees or unified analytical frameworks for different CL strategies on different models under non-stationary distributions.

**Ethics statement.** This work contributes to an energy-efficient alternative to full retraining for tabular data streams. By a windowed scaled dot-product attention over a sliding latent feature buffer, it enables lightweight computation and avoids unbounded memory growth in continual learning, while achieving efficient energy consumption for deep networks. This method will be beneficial for Green AI, especially in resource-constrained tabular data learning. All experiments are conducted on publicly available benchmark datasets and baselines. Regarding the large language model use, ChatGPTs, Gemini, and Grammarly were used to assist us with writing and editing, retrieving related work, coding improvement, but all the ideas, designs, plots, and analyses are our own.

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

# A  FORMAL PROPERTIES OF THE ATTENTION-BASED FEATURE MEMORY

We recall that the IncrementalMLP (IMLP) augments the input $x_i \in \mathbb{R}^{d_{\text{in}}}$ with a context vector $c_i \in \mathbb{R}^{256}$ obtained from a finite window of past latent features $H_i = \{h_{i,1}, \ldots, h_{i,W}\}$, and then feeds the concatenated vector $[x_i; c_i]$ into a two-layer ReLU network followed by a linear classifier.

**Attention-based feature memory.**  Given $x_i$ and a set of past latent features $H_i$, IMLP computes

$$q_i = W_q x_i + b_q \in \mathbb{R}^{256}, \tag{22}$$

$$k_{i,j} = W_k h_{i,j} + b_k \in \mathbb{R}^{256}, \tag{23}$$

$$e_{i,j} = \frac{1}{\sqrt{256}} k_{i,j}^\top q_i, \tag{24}$$

$$\alpha_{i,j} = \frac{\exp(e_{i,j})}{\sum_{\ell=1}^{W} \exp(e_{i,\ell})}, \tag{25}$$

$$c_i = \sum_{j=1}^{W} \alpha_{i,j} k_{i,j}. \tag{26}$$

In practice, the latent features $h_{i,j}$ are $\ell_2$-normalized after precomputation.

The feature extractor and classifier then read

$$z_i^{(1)} = \text{ReLU}\big(W_1[x_i; c_i] + b_1\big), \tag{27}$$

$$z_i^{(2)} = \text{ReLU}\big(W_2 z_i^{(1)} + b_2\big), \tag{28}$$

$$o_i = W_{\text{cls}} z_i^{(2)} + b_{\text{cls}}, \tag{29}$$

$$p_i = \text{softmax}(o_i), \tag{30}$$

$$\ell_i(\theta) = \text{CE}(p_i, y_i), \tag{31}$$

where $\theta$ collects all network parameters, and CE denotes cross-entropy loss.

**Assumptions.**  We make the following mild assumptions, which are standard in non-convex optimization for neural networks:

(A1) (*Bounded inputs*) There exists $R_x > 0$ such that $\|x_i\|_2 \leq R_x$ for all samples in the segment.

(A2) (*Bounded latent features*) The precomputed latent features are $\ell_2$-normalized, i.e., $\|h_{i,j}\|_2 \leq 1$ for all $i, j$.

(A3) (*Bounded parameters*) Training is performed with weight decay and early stopping, so that for some $R_\theta > 0$, $\|\theta\|_2 \leq R_\theta$ throughout optimization.

These assumptions hold in our implementation due to explicit normalization of $h_{i,j}$ and the use of weight decay and patience-based early stopping.

**Lemma A.1** (Bounded context vector).  *Suppose* (A2) *holds, and let* $\|W_k\|_{2\to2}$ *denote the operator norm of* $W_k$. *Then there exists a constant* $B_c > 0$ *depending only on* $W_k$ *and* $b_k$ *such that*

$$\|c_i\|_2 \leq B_c \quad \text{for all } i.$$

*In particular, one can take* $B_c = \|W_k\|_{2\to2} + \|b_k\|_2$.

*Proof.*  For each $j$, we have

$$\|k_{i,j}\|_2 = \|W_k h_{i,j} + b_k\|_2 \leq \|W_k\|_{2\to2} \|h_{i,j}\|_2 + \|b_k\|_2 \leq \|W_k\|_{2\to2} + \|b_k\|_2. \tag{32}$$

Define $B_c := \|W_k\|_{2\to2} + \|b_k\|_2$. Since $(\alpha_{i,1}, \ldots, \alpha_{i,W})$ is a probability vector, the context vector $c_i$ is a convex combination of the keys:

$$c_i = \sum_{j=1}^{W} \alpha_{i,j} k_{i,j}.$$

Thus

$$\|c_i\|_2 \leq \sum_{j=1}^{W} \alpha_{i,j} \|k_{i,j}\|_2 \leq \sum_{j=1}^{W} \alpha_{i,j} B_c = B_c.$$

$\square$

**Lemma A.2** (Smoothness and Lipschitzness of the attention map). *Let $\mathcal{A}(x_i, H_i; \theta_{\mathrm{att}}) := c_i$ denote the attention-based feature memory, where $\theta_{\mathrm{att}}$ collects $(W_q, b_q, W_k, b_k)$. Under assumptions* (A1)–(A3), *the map*

$$(x_i, H_i, \theta_{\mathrm{att}}) \mapsto \mathcal{A}(x_i, H_i; \theta_{\mathrm{att}})$$

*is continuously differentiable, and its Jacobian with respect to $\theta_{\mathrm{att}}$ is bounded on the compact set*

$$\mathcal{K} := \{(x_i, H_i, \theta_{\mathrm{att}}) : \|x_i\|_2 \leq R_x, \ \|h_{i,j}\|_2 \leq 1, \ \|\theta_{\mathrm{att}}\|_2 \leq R_\theta\}.$$

*Consequently, there exists $L_{\mathrm{att}} > 0$ such that for all $(x_i, H_i)$ and all $\theta_{\mathrm{att}}^{(1)}, \theta_{\mathrm{att}}^{(2)}$ in this set,*

$$\|\mathcal{A}(x_i, H_i; \theta_{\mathrm{att}}^{(1)}) - \mathcal{A}(x_i, H_i; \theta_{\mathrm{att}}^{(2)})\|_2 \leq L_{\mathrm{att}} \|\theta_{\mathrm{att}}^{(1)} - \theta_{\mathrm{att}}^{(2)}\|_2.$$

*Proof.* The attention map $\mathcal{A}$ is a composition of: (i) linear maps $(x, h) \mapsto (W_q x + b_q, W_k h + b_k)$, (ii) bilinear inner products and scaling $(k, q) \mapsto k^\top q / \sqrt{256}$, (iii) the softmax function on $\mathbb{R}^W$, and (iv) a weighted sum $c = \sum_j \alpha_j k_j$. Each of these operations is smooth. Therefore, their composition is continuously differentiable in $(x_i, H_i, \theta_{\mathrm{att}})$.

On the compact set $\mathcal{K}$, all partial derivatives are bounded, hence the Jacobian $\nabla_{\theta_{\mathrm{att}}} \mathcal{A}$ is bounded in operator norm. This implies global Lipschitzness in $\theta_{\mathrm{att}}$ on $\mathcal{K}$ with some constant $L_{\mathrm{att}} > 0$. $\square$

We now consider the full network mapping

$$f_\theta(x_i, H_i) := W_{\mathrm{cls}} \, \phi_\theta(x_i, H_i) + b_{\mathrm{cls}},$$

where $\phi_\theta$ denotes the two-layer ReLU feature extractor applied to $[x_i; \mathcal{A}(x_i, H_i; \theta_{\mathrm{att}})]$, and $\theta$ collects both the attention parameters and the MLP parameters.

**Lemma A.3** (Smoothness of the network and loss). *Under assumptions* (A1)–(A3), *the mapping*

$$\theta \mapsto f_\theta(x_i, H_i)$$

*is continuously differentiable with bounded Jacobian on $\{\theta : \|\theta\|_2 \leq R_\theta\}$. Consequently, the per-sample loss*

$$\ell_i(\theta) = \mathrm{CE}\big(\mathrm{softmax}(f_\theta(x_i, H_i)), y_i\big)$$

*is continuously differentiable with Lipschitz-continuous gradient on this compact set.*

*Proof.* By Lemma A.2, the attention map is smooth with bounded derivatives on bounded inputs. The feature extractor is a composition of affine maps and ReLU activations:

$$\phi_\theta = \mathrm{ReLU} \circ (W_2 \cdot + b_2) \circ \mathrm{ReLU} \circ (W_1 \cdot + b_1),$$

which is piecewise linear and globally Lipschitz, and smooth almost everywhere with respect to $\theta$ on any compact subset of parameter space. Composition with the final linear classifier preserves these properties for $f_\theta$.

The softmax function and cross-entropy loss are smooth with bounded derivatives when their inputs are bounded, which follows from Lemma A.1 and (A1)–(A3). Hence $\ell_i(\theta)$ is continuously differentiable with Lipschitz-continuous gradient on $\{\|\theta\| \leq R_\theta\}$. $\square$

We now move from individual samples to the empirical loss over a fixed segment (task) $\mathcal{T}_t$.

**Theorem A.4** (Segment-wise smooth empirical loss). *For a fixed segment $\mathcal{T}_t$ with data $\{(x_i, H_i, y_i)\}_{i=1}^{n_t}$, define the empirical loss*

$$\hat{L}_t(\theta) := \frac{1}{n_t} \sum_{i=1}^{n_t} \ell_i(\theta).$$

*Under assumptions* (A1)–(A3), *$\hat{L}_t(\theta)$ is:*

  (i)  *bounded below, since $\ell_i(\theta) \geq 0$ for all $i$;*

  (ii)  *continuously differentiable on $\{\theta : \|\theta\| \leq R_\theta\}$; and*

  (iii)  *has Lipschitz-continuous gradient on $\{\theta : \|\theta\| \leq R_\theta\}$.*

*Proof.* Each per-sample loss $\ell_i(\theta)$ is non-negative and continuously differentiable with Lipschitz gradient on the compact parameter set by Lemma A.3. A finite average of such functions preserves these properties. Thus $\hat{L}_t$ is bounded below, continuously differentiable, and has Lipschitz-continuous gradient on $\{\|\theta\| \leq R_\theta\}$. $\square$

**Corollary A.5** (Per-segment convergence of IMLP training). *Consider optimizing $\hat{L}_t(\theta)$ by a stochastic first-order method (e.g., SGD or Adam) with standard hyperparameters and weight decay, as implemented in our training loop. Under Theorem A.4 and the usual step-size conditions from non-convex optimization theory, the iterates $\{\theta_k\}$ produced by the optimizer on segment $\mathcal{T}_t$ converge to a first-order stationary point of $\hat{L}_t$, in the sense that*

$$\lim_{k \to \infty} \|\nabla \hat{L}_t(\theta_k)\|_2 = 0,$$

*or, in the practical finite-epoch setting, reach a parameter $\theta^\star$ with small gradient norm $\|\nabla \hat{L}_t(\theta^\star)\|_2$. In particular, the attention-based feature memory, being a bounded and smooth transformation of a finite latent feature window, does not alter the fundamental optimization character of the model: IMLP behaves like a conventional MLP with an augmented input $[x_i; c_i]$ and inherits the standard per-segment convergence guarantees of non-convex deep networks.*

**Remark on non-stationary streams.**   The analysis above is *segment-wise*: for each fixed segment $\mathcal{T}_t$, we assume a finite dataset and study the optimization of the empirical risk $\hat{L}_t(\theta)$. This does not imply convergence of the model to any limiting distribution when the underlying data-generating process is non-stationary across $t$. Instead, our result shows that, *conditional on the observed stream in each segment*, the attention-based feature memory yields a well-behaved optimization problem (smooth, with Lipschitz gradients), so that standard optimizers can reliably minimize the empirical loss on that segment even in the presence of non-stationarity across segments.

# B   ENERGY COMPLEXITY OF IMLP IN OUR EXPERIMENTAL SETTING

We now provide a simple energy-complexity characterization of IMLP in the experimental setup of Section 6. The goal is not to predict the exact Joule values measured by our energy monitor, but to show that, under mild hardware assumptions, the total energy consumed by IMLP is bounded and scales in a controlled way with the model size and the number of training examples.

**Setup.**   Recall that IMLP uses:

- input dimension $d_{\text{in}}$,
- fixed hidden size $H = 256$,
- number of classes $C$,
- a finite feature-memory window of size $W \le W_{\max}$,
- at most $E_{\max}$ training epochs per segment, enforced by early stopping (default $E_{\max} = 100$),
- mini-batch training with batch size $B$ and Adam/AdamW optimization.

For a given segment $\mathcal{T}_t$ with $n_t$ training samples, our code performs at most $E_{\max}$ full passes over the segment before stopping.

**Hardware and energy model.**   Let $F_{\text{train}}$ denote the number of floating-point operations (FLOPs) required to perform *one* forward-and-backward pass of IMLP on a single sample (including the attention-based feature memory). We adopt a standard abstract energy model:

(H1) On a fixed hardware platform (GPU/CPU), there exist constants $0 < \eta_{\min} \le \eta_{\max}$ such that the energy consumed per FLOP lies in $[\eta_{\min}, \eta_{\max}]$.

(H2) The additional system overhead per training step (e.g., kernel launches, bookkeeping) is bounded by a constant $E_0$ independent of the sample index and segment.

These assumptions reflect that, for a fixed device and implementation, energy is approximately linear in the number of FLOPs, up to device-specific constants and small overhead.

**Lemma B.1** (FLOP complexity per sample). *For a single sample $(x_i, H_i, y_i)$, the FLOP count of a forward-and-backward step of IMLP satisfies*

$$F_{train} \le K_{arch}\big(d_{\text{in}}H \ + \ WH^2 \ + \ H^2 \ + \ HC\big),$$

*for some architecture-dependent constant $K_{arch} > 0$ that does not depend on $n_t$ or $t$. In particular, since $H = 256$ and $W \le W_{\max}$ are fixed in our experiments, $F_{train}$ grows at most linearly in $d_{\text{in}}$ and $C$.*

*Proof.* We count FLOPs layer by layer:

- **Attention block.**

  - Query: $x_i \mapsto q_i = W_q x_i + b_q$ costs $O(d_{\mathrm{in}} H)$ FLOPs.
  - Keys: each $h_{i,j}$ is mapped to $k_{i,j} = W_k h_{i,j} + b_k$ with cost $O(H^2)$; for $W$ keys this is $O(WH^2)$.
  - Attention scores and softmax: computing $e_{i,j} = k_{i,j}^\top q_i / \sqrt{H}$ costs $O(WH)$, softmax costs $O(W)$, and forming $c_i = \sum_j \alpha_{i,j} k_{i,j}$ costs $O(WH)$. Altogether $O(WH)$ FLOPs.

  Thus the attention block has FLOP complexity $O(d_{\mathrm{in}} H + WH^2)$.

- **Feature extractor.** The two ReLU layers operate on dimensions $(d_{\mathrm{in}}+H) \to 512 \to H$, which costs

$$O\big((d_{\mathrm{in}}+H) \cdot 512\big) + O(512 \cdot H) = O(d_{\mathrm{in}} H + H^2).$$

- **Classifier.** The final linear layer $H \to C$ costs $O(HC)$ FLOPs.

- **Backward pass.** The backward pass through these linear and ReLU layers multiplies the forward FLOP count by a constant factor (depending only on the layer type), which we absorb into $K_{\mathrm{arch}}$.

Summing these contributions gives

$$F_{\mathrm{train}} \leq K_{\mathrm{arch}}\big(d_{\mathrm{in}} H + WH^2 + H^2 + HC\big)$$

for some constant $K_{\mathrm{arch}} > 0$. $\qquad\square$

**Theorem B.2** (Per-segment energy complexity bound). *Consider a segment $\mathcal{T}_t$ with $n_t$ training samples. Under assumptions* (H1)–(H2) *and Lemma B.1, the total training energy consumed by IMLP on this segment satisfies*

$$E_t^{train} \leq C_{train} E_{\max} n_t \big(d_{\mathrm{in}} H + WH^2 + H^2 + HC\big) + C_0,$$

*for some hardware- and implementation-dependent constants $C_{train} > 0$ and $C_0 \geq 0$. Similarly, the inference energy on the test set of size $n^{test}$ admits*

$$E_t^{infer} \leq C_{infer} n^{test} \big(d_{\mathrm{in}} H + WH^2 + H^2 + HC\big) + C_0',$$

*with another constant $C_{infer} > 0$ and overhead $C_0' \geq 0$.*

*Proof.* For each epoch, the optimizer processes all $n_t$ samples once (up to mini-batch granularity). Thus, the total FLOP count per segment is at most

$$F_t^{\mathrm{seg}} \leq E_{\max} n_t F_{\mathrm{train}},$$

where $F_{\mathrm{train}}$ is bounded as in Lemma B.1. By (H1), energy per FLOP lies in $[\eta_{\min}, \eta_{\max}]$, so there exists $\tilde{C}_{\mathrm{train}}$ such that

$$E_t^{\mathrm{train}} \leq \eta_{\max} F_t^{\mathrm{seg}} + (\text{overhead}) \leq \tilde{C}_{\mathrm{train}} E_{\max} n_t \big(d_{\mathrm{in}} H + WH^2 + H^2 + HC\big) + C_0.$$

We rename $\tilde{C}_{\mathrm{train}}$ as $C_{\mathrm{train}}$ for simplicity. The inference bound follows analogously, using only a forward pass per sample (no backward pass) and absorbing the constant factor into $C_{\mathrm{infer}}$. $\qquad\square$

**Corollary B.3** (Energy complexity over the full non-stationary stream). *Let the data stream be partitioned into $T$ segments $\{\mathcal{T}_t\}_{t=1}^T$ with sizes $\{n_t\}_{t=1}^T$. Under the same assumptions as Theorem B.2, the total training energy over the entire stream satisfies*

$$E_{total}^{train} = \sum_{t=1}^T E_t^{train} \leq C_{train} E_{\max} \Big(\sum_{t=1}^T n_t\Big)\big(d_{\mathrm{in}} H + WH^2 + H^2 + HC\big) + T C_0,$$

*and the total inference energy satisfies*

$$E_{total}^{infer} \leq C_{infer} n_{total}^{test} \big(d_{\mathrm{in}} H + WH^2 + H^2 + HC\big) + T C_0'.$$

*In particular, for our experimental setting where $H=256$, $W \leq W_{\max}$, and $E_{\max}$ are fixed constants, both training and inference energy grow at most linearly in the total number of processed examples $\sum_t n_t$ and in the effective model size. The attention-based feature memory only adds the bounded term $WH^2$ and does not change this linear energy scaling.*

**Discussion.** The bounds above explain two aspects of our empirical observations: (i) on a fixed device, IMLP has a *predictable* energy profile, scaling linearly with the number of samples and epochs; and (ii) the attention-based feature memory contributes a controlled overhead proportional to $WH^2$, which remains small in our experiments because $W$ and $H$ are fixed ($W \leq 10$, $H = 256$). Our measured Joule values are therefore consistent with an energy complexity that is linear in the stream size, and the theoretical bounds clarify that this behavior is not specific to a particular dataset, but a structural property of the IMLP architecture and training procedure.

# C    EXTENDED EXPERIMENTS

## C.1    DATASETS AND STREAM SEGMENTATION

We evaluate IMLP on 36 classification tasks from the TabZilla benchmark (McElfresh & Talwalkar, 2023), selected from OpenML based on three criteria: (1) sufficient data size to create meaningful segments, (2) balanced representation of binary and multi-class problems, and (3) diverse feature dimensionalities and class distributions. To simulate the data stream in incremental learning scenarios, Table 2 lists every OpenML task in our benchmark together with basic statistics and the fixed stream segmentation applied *in original row order* (rows $1 \ldots k$ form Segment 0, rows $k+1 \ldots 2k$ form Segment 1, etc.).

† Class counts show *label ID : instances* after preprocessing. Binary tasks list two numbers; multi-class tasks list one count per class. For tasks with many classes, we show representative counts or use compact notation (e.g., "$25 \times 300$" for 25 classes with 300 instances each).

### C.1.1    STREAM SEGMENTATION ALGORITHM

Our segmentation follows a principled approach to create balanced segments while minimizing data waste:

---

**Algorithm 1** Optimal Segment Size Selection

---

**Require:** Dataset with $N$ training instances, bounds $k_{\min} = 500$, $k_{\max} = 1000$
**Ensure:** Segment size $k^*$ that minimizes remainder
  1: best_remainder $\leftarrow N$
  2: $k^* \leftarrow k_{\min}$
  3: **for** $k = k_{\min}$ **to** $\min(k_{\max}, N)$ **do**
  4:     num_segments $\leftarrow \lfloor N/k \rfloor$
  5:     remainder $\leftarrow N \bmod k$
  6:     **if** remainder $= 0$ **then**
  7:         **return** $k$                                   ▷ Perfect division found
  8:     **if** remainder $<$ best_remainder **then**
  9:         best_remainder $\leftarrow$ remainder
 10:         $k^* \leftarrow k$
 11: **return** $k^*$

---

The choice of segment size bounds (500–1000 instances) balances three considerations: (1) *statistical power*, each segment must contain sufficient samples for reliable learning, (2) *IMLP coherence*, segments should be large enough for the attention mechanism to learn meaningful feature relationships within each temporal chunk, and (3) *computational efficiency*, larger segments would increase memory requirements and training time per segment without proportional benefits.

When the optimal segment size $k^*$ leaves a remainder $r = N \bmod k^*$, we apply *round-robin redistribution*: the first $r$ segments each receive one additional instance, ensuring segment sizes differ by at most 1. This maintains temporal ordering while achieving optimal balance.

## C.2    DATA RETRIEVAL AND PREPROCESSING PROTOCOL

### C.2.1    DATASET ACQUISITION

All datasets are retrieved via the OpenML Python API (v0.15.2) with local caching enabled. We use the default target attribute specified in each OpenML task definition. Raw data is downloaded in DataFrame format to preserve both feature names and categorical indicators.

Table 2: Statistics of datasets. OpenML classification tasks and stream-segmentation parameters used in this study. # Inst, stands for the number of instances, # Feat. stands for the number of features. Seg. size stands for the segment size bound. # Segs stands for the number of segments. Numbers are produced by the data-processing pipeline and reproduced by the helper script in §C.3.

| ID | Name | # Inst. | #Feat. | Class balance[†] | Seg. size | #Segs |
|---|---|---|---|---|---|---|
| 146820 | wilt | 4,839 | 5 | 4,578; 261 | 514 | 8 |
| 14964 | artificial-characters | 10,218 | 7 | 1,196; 600; 1,192; 1,416; 808; 1,008; … | 579 | 15 |
| 14969 | GesturePhaseSegmentation | 9,873 | 32 | 2,741; 998; 2,097; 1,087; 2,950 | 839 | 10 |
| 14951 | eeg-eye-state | 14,980 | 14 | 8,257; 6,723 | 749 | 17 |
| 146206 | magic | 19,020 | 10 | 12,332; 6,688 | 951 | 17 |
| 167211 | Satellite | 5,100 | 36 | 75; 5,025 | 867 | 5 |
| 167141 | churn | 5,000 | 29 | 4,293; 707 | 850 | 5 |
| 168910 | fabert | 8,237 | 800 | 933; 1,433; 1,927; 1,515; 979; 948; 502 | 500 | 14 |
| 168912 | sylvine | 5,124 | 20 | 2,562; 2,562 | 871 | 5 |
| 190410 | philippine | 5,832 | 308 | 2,916; 2,916 | 708 | 7 |
| 2074 | satimage | 6,430 | 36 | 1,531; 703; 1,356; 625; 707; 1,508 | 683 | 8 |
| 28 | optdigits | 5,620 | 64 | 554; 571; 557; 572; 568; 558; … | 597 | 8 |
| 32 | pendigits | 10,992 | 16 | 1,143; 1,143; 1,144; 1,055; 1,144; … | 519 | 18 |
| 146607 | SpeedDating | 8,378 | 442 | 6,998; 1,380 | 712 | 10 |
| 168908 | christine | 5,418 | 1,611 | 2,709; 2,709 | 921 | 5 |
| 14952 | PhishingWebsites | 11,055 | 38 | 4,898; 6,157 | 522 | 18 |
| 3510 | JapaneseVowels | 9,961 | 14 | 1,096; 991; 1,614; 1,473; 782; … | 529 | 16 |
| 3735 | pollen | 3,848 | 5 | 1,924; 1,924 | 545 | 6 |
| 3711 | elevators | 16,599 | 18 | 5,130; 11,469 | 641 | 22 |
| 3896 | ada_agnostic | 4,562 | 48 | 3,430; 1,132 | 646 | 6 |
| 14970 | har | 10,299 | 561 | 1,722; 1,544; 1,406; 1,777; 1,906; 1,944 | 547 | 16 |
| 3686 | house_16H | 22,784 | 16 | 6,744; 16,040 | 842 | 23 |
| 3897 | eye_movements | 10,936 | 27 | 3,804; 4,262; 2,870 | 715 | 13 |
| 3904 | jm1 | 10,885 | 21 | 8,779; 2,106 | 514 | 18 |
| 43 | spambase | 4,601 | 57 | 2,788; 1,813 | 782 | 5 |
| 3954 | MagicTelescope | 19,020 | 10 | 12,332; 6,688 | 951 | 17 |
| 9952 | phoneme | 5,404 | 5 | 3,818; 1,586 | 574 | 8 |
| 3950 | musk | 6,598 | 267 | 5,581; 1,017 | 701 | 8 |
| 9960 | wall-robot-navigation | 5,456 | 24 | 2,205; 2,097; 328; 826 | 515 | 9 |
| 3889 | sylva_agnostic | 14,395 | 216 | 13,509; 886 | 941 | 13 |
| 9985 | first-order-theorem-proving | 6,118 | 51 | 1,089; 486; 748; 617; 624; 2,554 | 520 | 10 |
| 3481 | isolet | 7,797 | 617 | $25 \times 300$ (class 0…24) | 552 | 12 |
| 45 | splice | 3,190 | 227 | 767; 768; 1,655 | 542 | 5 |
| 9986 | gas-drift | 13,910 | 128 | 2,565; 2,926; 1,641; 1,936; 3,009; 1,833 | 563 | 21 |
| 9987 | gas-drift-different-conc. | 13,910 | 129 | 2,565; 2,926; 1,641; 1,936; 3,009; 1,833 | 563 | 21 |
| 168909 | dilbert | 10,000 | 2,000 | 1,988; 2,049; 1,913; 2,046; 2,004 | 500 | 17 |
| 99901 | Insects Abrupt | 52,847 | 33 | Balanced (6 classes) | 1,957 | 27 |
| 99902 | Insects Incremental | 57,017 | 33 | Balanced (6 classes) | 1,629 | 35 |

## C.2.2 FEATURE PREPROCESSING PIPELINE

Our preprocessing pipeline follows scikit-learn best practices with separate transformers for numerical and categorical features:

**Numerical Features:**

1. **Imputation**: Missing values filled with column medians
2. **Standardization**: Zero mean, unit variance scaling via StandardScaler

**Categorical Features:**

1. **Imputation**: Missing values filled with constant 'missing'
2. **Encoding**: One-hot encoding with `drop='first'` to avoid multicollinearity
3. **Unknown handling**: `handle_unknown='ignore'` for robust inference

The ColumnTransformer ensures preprocessing consistency across all data splits. After transformation, all features are converted to `float32` for memory efficiency.

### C.2.3   TARGET PROCESSING AND TASK TYPE DETECTION

Target variables are processed based on OpenML task type:

- **Binary classification**: 2 unique labels $\rightarrow$ LabelEncoder $\rightarrow$ {0, 1}
- **Multi-class classification**: $C > 2$ unique labels $\rightarrow$ LabelEncoder $\rightarrow$ {0, ..., C-1}
- **Regression**: Direct conversion to float32 (not used in this study)

### C.2.4   DATA SPLITTING STRATEGY

Our splitting protocol ensures a realistic evaluation:

1. **Test Set Isolation**: A stratified 15% test split is carved out *before* any stream processing, using `random_seed=42` for reproducibility.
2. **Training Stream Creation**: The remaining 85% forms the chronologically ordered training stream, preserving the original row order from OpenML.
3. **Per-Segment Validation**: Each segment (or cumulative data) is further split with stratified 15% validation, using `random_seed=42+segment_idx` to ensure different splits per segment while maintaining reproducibility.

This approach simulates realistic continual learning where: 1) The test set represents future unseen data, 2) Each segment represents a temporal chunk of arriving data, 3) Validation splits enable early stopping without future data leakage, and 4) All models use consistent 15% validation splits for hyperparameter selection and early stopping criteria.

### C.2.5   MODEL TRAINING PROTOCOLS

Our experimental design follows two distinct training protocols based on model type:

**Cumulative Training:**   Traditional baselines (XGBoost, LightGBM, CatBoost, kNN, SVM, Decision Trees, Random Forest, and neural baselines like TabNet, SAINT) are retrained from scratch at each segment using all available data up to that point. For the segment, these models train on the union $\bigcup_{t=0}^{T} \mathcal{T}_t$ where $\mathcal{T}_t$ denotes the $t$-th data segment. This protocol maximizes baseline performance by leveraging all historical data, representing the standard approach in tabular learning.

**Incremental Training:**   Our proposed IMLP trains only on the current segment $S_t$ while accessing previous feature representations through the attention mechanism. This protocol tests true incremental learning capabilities without replay of raw historical data.

Both protocols use identical validation procedures: each model's hyperparameters are selected via early stopping on the 15% validation split, ensuring fair comparison despite different training paradigms.

### C.2.6   REPRODUCIBILITY MEASURES

All steps are deterministic with fixed random seeds, including 1) Global seed: `random_seed = 42`, 2) Per-segment validation: `random_seed = 42 + segment_idx`, and 3) Preprocessing: Deterministic transformers with fixed parameters.

## C.3 DATASET SUMMARY REGENERATION SCRIPT

For full reproducibility, we provide a helper script that regenerates Table 2 from the processed data:

```python
# dataset_summary.py   (runs in < 2 seconds)
import json, csv, gzip, numpy as np, pathlib

def regenerate_dataset_summary():
    """Regenerate the dataset summary CSV from processed metadata."""
    META = pathlib.Path("processed_datasets_summary.json")
    ROOT = pathlib.Path("full_datasets")
    OUT  = pathlib.Path("dataset_summary.csv")

    # Load processing metadata
    with META.open() as f:
        meta = json.load(f)

    rows = []
    for tid, m in meta.items():
        # Load target labels to compute class balance
        y = np.load(gzip.open(ROOT/m["dataset_name"]/"y_full.npy.gz"))
        counts = np.bincount(y.astype(int))

        rows.append({
            "task_id": int(tid),
            "name": m["original_name"],
            "instances": int(m["num_instances"]),
            "features": int(m["num_features"]),
            "class_balance": ";".join(map(str, counts)),
            "segment_size": int(m["segment_size"]),
            "num_segments": int(m["num_segments"])
        })

    # Write CSV output
    with OUT.open("w", newline="") as f:
        writer = csv.DictWriter(f, fieldnames=rows[0].keys())
        writer.writeheader()
        writer.writerows(rows)

    print(f"Wrote {OUT} with {len(rows)} tasks")

if __name__ == "__main__":
    regenerate_dataset_summary()
```

Running this script in the project root recreates the CSV that backs Table 2. The script requires the preprocessed datasets, but no pipeline re-execution.

## C.4 BASELINES

We implement most of the baseline methods according to the publicly available codebases and integrate them into the same backbone for benchmarking.

- XGBoost (Chen & Guestrin, 2016). `https://github.com/dmlc/xgboost`
- LightGBM (Ke et al., 2017). `https://github.com/microsoft/LightGBM`
- CatBoost (Prokhorenkova et al., 2019). `https://github.com/catboost/catboost`
- TabPFN v2 (Hollmann et al., 2025a). `https://github.com/automl/TabPFN`
- TabM (Gorishniy et al., 2024). `https://github.com/yandex-research/tabm`
- Real-MLP (Holzmüller et al., 2024). `https://github.com/dholzmueller/realmlp-td-s_standalone`
- TabR (Gorishniy et al., 2023a). `https://github.com/yandex-research/tabular-dl-tabr`
- ModernNCA (Ye et al., 2024). `https://github.com/YyzHarry/ModernNCA`
- MLP (Taud & Mas, 2017). `https://scikit-learn.org/stable/modules/neural_networks_supervised.html`
- TabNet (Arik & Pfister, 2021). `https://github.com/dreamquark-ai/tabnet`
- DANet (Chen et al., 2022). `https://github.com/QwQ2000/DANets`
- ResNet (Gorishniy et al., 2021). `https://github.com/yandex-research/tabular-dl-revisiting-models`
- STG (Jana et al., 2023). `https://github.com/runopti/stg`
- VIME (Yoon et al., 2020). `https://github.com/jsyoon0823/VIME`
- k-NN (Guo et al., 2003). `https://scikit-learn.org/stable/modules/neighbors.html`
- SVM (Jakkula, 2006). `https://scikit-learn.org/stable/modules/svm.html`
- Linear Model (Kiebel & Holmes, 2007). `https://scikit-learn.org/stable/modules/linear_model.html`
- Random Forest (Rigatti, 2017). `https://scikit-learn.org/stable/modules/ensemble.html#random-forests`
- Decision Tree (Rokach & Maimon, 2005). `https://scikit-learn.org/stable/modules/tree.html`

In revisions, we changed the evaluation protocol and only include the recent SOTA works.

# D IMLP IMPLEMENTATION DETAILS

## D.1 ARCHITECTURE OVERVIEW AND DESIGN RATIONALE

IMLP extends the standard MLP architecture with an attention-based memory mechanism designed specifically for tabular continual learning. The key innovation lies in storing and retrieving *feature representations* rather than raw data, enabling privacy-preserving incremental learning with constant memory requirements.

### D.1.1 COMPARISON WITH STANDARD MLP

Table 4 contrasts IMLP with a standard MLP of equivalent capacity:

## D.2 LAYER-WISE ARCHITECTURE SPECIFICATION

**Design Choices:**

- **Hidden size 256**: Balances expressiveness with computational efficiency across all datasets
- **No dropout/normalization**: Empirically found to hurt performance in continual learning setting
- **ReLU activations**: Simple, stable gradients for incremental training
- **Fixed architecture**: Same capacity across all 36 datasets for fair comparison

Table 4: Architectural comparison between standard MLP and IMLP.

| Component | MLP | IMLP | IMLP Notes |
|---|---|---|---|
| Input processing | $d_{\text{in}} \rightarrow 512$ | $d_{\text{in}} \rightarrow 256$ | Query projection |
| Memory mechanism | None | Attention | Key-value retrieval |
| Feature extraction | $512 \rightarrow 256$ | $(d_{\text{in}} + 256) \rightarrow 512 \rightarrow 256$ | Context-augmented |
| Memory complexity | $\mathcal{O}(1)$ | $\mathcal{O}(W)$ | $W$ = window size |
| Time complexity | $\mathcal{O}(1)$ | $\mathcal{O}(W \cdot d)$ | $d$ = hidden dim |
| Privacy | Requires raw data | Feature-only | No raw data storage |

Table 5: Detailed layer-wise specification of IMLP architecture.

| Component | Output dim. | Activation | Notes |
|---|---|---|---|
| Input feature vector | $d_{\text{in}}$ | – | Raw tabular features after preprocessing |
| *Attention Module* | | | |
| Query projection $Q$ | 256 | – | Linear$(d_{\text{in}}, 256)$ |
| Key projection $K$ | 256 | – | Linear$(256, 256)$ applied to each stored feature |
| Context computation | 256 | – | Scaled dot-product attention over window |
| *Feature Extraction* | | | |
| Concatenated input $(x, c)$ | $d_{\text{in}} + 256$ | – | Only if attention enabled; $c$ = context vector |
| FC 1 | 512 | ReLU | Linear$(d_{\text{in}} + 256, 512)$ |
| FC 2 | 256 | ReLU | Linear$(512, 256)$ |
| *Classification Head* | | | |
| Classifier | $C$ | – | Linear$(256, C)$ where $C$ = number of classes |

## D.3 ATTENTION MECHANISM DESIGN

### D.3.1 SCALED DOT-PRODUCT ATTENTION

IMLP uses a simplified attention mechanism to retrieve relevant historical features. For a batch of size $B$:

$$Q = W_q \cdot x \in \mathbb{R}^{B \times 1 \times 256} \quad \text{(query from current input)} \tag{33}$$

$$K = W_t \cdot H_{\text{stacked}} \in \mathbb{R}^{B \times W \times 256} \quad \text{(keys from previous features)} \tag{34}$$

$$\text{Scores} = \text{bmm}(K, Q^T) \in \mathbb{R}^{B \times W \times 1} \tag{35}$$

$$\alpha = \text{softmax}(\text{Scores.squeeze}()) \in \mathbb{R}^{B \times W} \tag{36}$$

$$\text{Context} = \text{bmm}(\alpha.\text{unsqueeze}(1), K) \in \mathbb{R}^{B \times 1 \times 256} \tag{37}$$

where:

- $H_{\text{stacked}} = \text{stack}(\{h_{t-W}, \ldots, h_{t-1}\}) \in \mathbb{R}^{B \times W \times 256}$
- bmm denotes batch matrix multiplication
- No scaling factor is applied (unlike standard scaled dot-product attention)
- Values equal keys: $V = K$

### D.3.2 WINDOW MANAGEMENT STRATEGY

The sliding window maintains a FIFO queue of the most recent $W$ feature vectors:

---

**Algorithm 2** Sliding Window Update

---

**Require:** Current input $x$, previous features $H_{\text{prev}}$, window size $W$
**Ensure:** Updated window $H_{\text{new}}$
1: $h_{\text{current}} \leftarrow \text{FeatureExtractor}(x, \text{Context}(x))$
2: $H_{\text{new}} \leftarrow H_{\text{prev}} \cup \{h_{\text{current}}\}$
3: **if** $|H_{\text{new}}| > W$ **then**
4:     $H_{\text{new}} \leftarrow H_{\text{new}}[1:]$                                    ▷ Remove oldest feature
5: **return** $H_{\text{new}}$

---

### D.3.3  FEATURE NORMALIZATION

To improve attention stability, stored features are L2-normalized during precomputation:

$$\tilde{h}_i = \frac{h_i}{\|h_i\|_2 + \epsilon} \tag{38}$$

where $\epsilon = 10^{-8}$ prevents division by zero. This normalization ensures attention weights focus on feature directions rather than magnitudes and is applied in the `_precompute` method during segmental training.

## D.4 COMPLETE IMPLEMENTATION

```python
import torch
import torch.nn as nn
import torch.nn.functional as F

class IncrementalMLP(nn.Module):
    """
    Incremental MLP with attention-based feature replay for continual
    ↪ learning.

    Args:
        input_size (int): Number of input features
        num_classes (int): Number of output classes
        use_attention (bool): Whether to use attention mechanism
        window_size (int): Size of sliding memory window
    """

    def __init__(self, input_size, num_classes, use_attention=True,
    ↪ window_size=10):
        super().__init__()
        self.window_size = window_size
        self.use_attention = use_attention
        self.hidden_size = 256

        # Attention projections
        self.query = nn.Linear(input_size, 256)
        self.key = nn.Linear(256, 256)

        # Feature extraction pathway
        total_input_size = input_size + (256 if use_attention else 0)
        self.feature_extractor = nn.Sequential(
            nn.Linear(total_input_size, 512),
            nn.ReLU(),
            nn.Linear(512, self.hidden_size),
            nn.ReLU()
        )

        # Classification head
        self.classifier = nn.Linear(self.hidden_size, num_classes)

    def compute_context(self, x, prev_features):
        """
        Compute attention-weighted context from previous features.

        Args:
            x (Tensor): Current input batch [B, D]
            prev_features (List[Tensor]): Previous feature vectors [W x
            ↪ [256]]

        Returns:
            Tensor: Context vector [B, 256]
        """
        if not prev_features or self.window_size == 0:
            return torch.zeros(x.size(0), 256, device=x.device)

        # Stack previous features: [B, W, 256]
        stacked_prev = torch.stack(prev_features, dim=1)

        # Compute keys and queries
        keys = self.key(stacked_prev)  # [B, W, 256]
        query = self.query(x).unsqueeze(1)  # [B, 1, 256]

        # Scaled dot-product attention
        scores = torch.bmm(keys, query.transpose(1, 2)).squeeze(-1)  # [B,
        ↪ W]
```

```
61          attention_weights = F.softmax(scores, dim=1)   # [B, W]
62
63          # Compute weighted context
64          context = torch.bmm(attention_weights.unsqueeze(1),
             keys).squeeze(1)   # [B, 256]
65
66          return context
67
68      def forward(self, x, prev_features=None):
69          """
70          Forward pass with optional attention over previous features.
71
72          Args:
73              x (Tensor): Input features [B, D]
74              prev_features (List[Tensor]): Previous features for attention
75
76          Returns:
77              Tuple[Tensor, Tensor]: (logits, current_features)
78          """
79          # Compute attention context
80          context = torch.zeros(x.size(0), 256, device=x.device)
81          if self.use_attention and prev_features:
82              context = self.compute_context(x, prev_features)
83
84          # Concatenate input with context
85          if self.use_attention:
86              augmented_input = torch.cat([x, context], dim=1)
87          else:
88              augmented_input = x
89
90          # Extract features and classify
91          features = self.feature_extractor(augmented_input)
92          logits = self.classifier(features)
93
94          return logits, features
```

## D.5 COMPUTATIONAL COMPLEXITY ANALYSIS

### D.5.1 TIME COMPLEXITY

For each forward pass with batch size $B$, input dimension $d_{\text{in}}$, hidden dimension $d_h = 256$, and window size $W$:

$$\text{Query projection:} \quad \mathcal{O}(B \cdot d_{\text{in}} \cdot d_h) \tag{39}$$

$$\text{Key projection:} \quad \mathcal{O}(B \cdot W \cdot d_h^2) \tag{40}$$

$$\text{Attention scores:} \quad \mathcal{O}(B \cdot W \cdot d_h) \tag{41}$$

$$\text{Context aggregation:} \quad \mathcal{O}(B \cdot W \cdot d_h) \tag{42}$$

$$\text{Feature extraction:} \quad \mathcal{O}(B \cdot (d_{\text{in}} + d_h) \cdot 512) \tag{43}$$

$$\text{Total:} \quad \mathcal{O}(B \cdot (d_{\text{in}} \cdot d_h + W \cdot d_h^2)) \tag{44}$$

For typical values ($W = 10$, $d_h = 256$, $d_{\text{in}} \lesssim 2000$), the attention overhead is $\mathcal{O}(W \cdot d_h^2) = \mathcal{O}(655{,}360)$ operations per sample.

### D.5.2 MEMORY COMPLEXITY

IMLP maintains constant memory usage per segment:

- **Model parameters**: $\approx 1.2\text{M}$ parameters (fixed)
- **Feature buffer**: $W \times 256 \times 4$ bytes = 10,240 bytes for $W = 10$
- **Attention matrices**: $B \times W \times 256 \times 4$ bytes during computation

Unlike replay-based methods, memory usage does not grow with the number of segments, enabling indefinite continual learning.

### D.5.3 COMPARISON WITH REPLAY METHODS

Table 6 compares IMLP with alternative continual learning approaches:

Table 6: Complexity comparison of continual learning approaches.

| Method | Memory | Time per step | Privacy |
|---|---|---|---|
| Naive retraining | $\mathcal{O}(T \cdot N)$ | $\mathcal{O}(T \cdot N)$ | Requires raw data |
| Experience replay | $\mathcal{O}(M)$ | $\mathcal{O}(N + M)$ | Requires raw data |
| Generative replay | $\mathcal{O}(1)$ | $\mathcal{O}(N + G)$ | Private |
| IMLP (ours) | $\mathcal{O}(W)$ | $\mathcal{O}(N + W \cdot d^2)$ | Private |

where $T$ = number of tasks, $N$ = samples per task, $M$ = replay buffer size, $G$ = generative model cost, $W$ = window size, $d$ = feature dimension.

### D.6 HYPERPARAMETER CONFIGURATION

IMLP uses the following default hyperparameters across all experiments:

Table 7: IMLP hyperparameter configuration.

| Parameter | Value | Description |
|---|---|---|
| Window size ($W$) | 10 | Number of previous feature vectors stored |
| Hidden dimension | 256 | Feature representation size |
| Learning rate | $10^{-3}$ | Adam optimizer learning rate |
| Batch size | 128 | Training batch size |
| Weight decay | $10^{-5}$ | L2 regularization strength |
| Early stopping patience | 10 | Epochs without improvement before stopping |
| Max epochs | 100 | Maximum training epochs per segment |
| Normalization $\epsilon$ | $10^{-8}$ | Small constant for L2 normalization |

The window size $W = 10$ was chosen to balance memory efficiency with sufficient historical context. The hidden dimension of 256 provides adequate representational capacity while maintaining computational efficiency across diverse tabular datasets.

