# OpenReview forum: "IMLP: An Energy-Efficient Continual Learning Method for Tabular Data Streams"
_ICLR.cc/2026/Conference — ICLR 2026 Conference Desk Rejected Submission_

### Official Review · Reviewer_kJho · 2025-10-25

**Soundness:** 3
**Presentation:** 2
**Contribution:** 3
**Rating:** 4
**Confidence:** 4

**Summary:**

This paper aims to address an important and practical problem in the field of continual learning for tabular data streams: how to achieve energy-efficient learning on resource-constrained devices. The authors propose an incremental Multi-Layer Perceptron (IMLP) model that utilizes a fixed-size sliding window storing latent features, combined with an attention mechanism, to avoid storing raw data and prevent unbounded memory overhead. Furthermore, the authors introduce a new evaluation metric, NetScore-T, to quantify the trade-off between a model's accuracy and its energy consumption.

**Strengths:**

1.  The task setting is very interesting.
2. The data processing and experimental details are thoroughly described in the appendix, which enhances the reproducibility and credibility of the study. In addition, the experimental metrics are comprehensively designed, and the comparison methods cover both recent popular paradigms and classical models in tabular learning, providing a solid basis for the objectivity of the results.

**Weaknesses:**

1. The paper's core method is a combination of existing, mature techniques such as sliding windows, attention mechanisms, and latent replay, rather than proposing a fundamentally new algorithm or theory. Therefore, its innovative academic contribution is limited.
2. Comparing the energy efficiency of the incrementally-trained IMLP against baseline models forced into a cumulative retraining setting leads to an obvious conclusion (that incremental updates are more efficient than retraining). This comparison fails to effectively demonstrate IMLP's superiority over other **continual learning** algorithms.
3. The paper fails to quantify the actual contribution of its key components (e.g., the attention mechanism) or analyze the impact of core hyperparameters (e.g., window size) through ablation studies, leaving the effectiveness and rationale of the model's design unverified.
4. Simulating a data stream by partitioning a static dataset ignores complex, real-world dynamics such as "gradual concept drift." This makes it questionable whether the model's performance in controlled experiments can be generalized to practical applications.
5. The caption in figure2 is too small.
6. As the authors mention at the beginning of the experimental section, most baseline models are not designed for streaming learning. To ensure a fair comparison, a “best-effort comparison” strategy is adopted. However, under this setting, IMLP performs incremental updates, while other models need to be retrained from scratch for each segment, which may introduce bias in the energy consumption results. Is such a comparison truly fair?
7. A strong motivation of this paper is to maintain comparable accuracy while improving energy efficiency. However, could the FIFO mechanism cause highly discriminative early features to be discarded too early, leading to catastrophic forgetting and affecting the model’s long-term stability? It would be helpful to include comparison experiments with different buffer strategies.

**Questions:**

Overall, I think this is a very interesting topic, but current version needs to be polished.

---

> ### Author Response · Authors · 2025-11-21
> **Response by Authors to Reviewer kJho (Weaknesses #1, #2, and #3)**
>
> #### Weaknesses
> ##### W1. The paper's core method is a combination of existing, mature techniques such as sliding windows, attention mechanisms, and latent replay, rather than proposing a fundamentally new algorithm or theory. Therefore, its innovative academic contribution is limited.
>
> [AW1] Admittedly, our IMLP model incorporates existing mature techniques. However, our design is rational and can be rigorously justified on theoretical grounds, not only a simple combination.
>
> We theoretically prove that we can guide the energy-accuracy trade-off with the segment-wise convergence of the proposed attention-based feature memory and the energy-complexity characterization of IMLP. Please kindly refer to [AW1] response by authors to reviewer xd8r and [AQ1] responses by authors to reviewer vdbK, respectively. To the best of our knowledge, this is the first approach to energy-efficient continual learning on tabular data streams at the time of this work.
>
> ##### W2. Comparing the energy efficiency of the incrementally-trained IMLP against baseline models forced into a cumulative retraining setting leads to an obvious conclusion (that incremental updates are more efficient than retraining). This comparison fails to effectively demonstrate IMLP's superiority over other continual learning algorithms.
>
> [AW2] We appreciate the suggestion to include more continual learning baselines. As discussed in the related works, several methods incorporate CL strategies with NNs. They include regularization-based approaches (EWC, SI, MAS, LwF), replay-based strategies (iCaRL and generative replay), attention-based retrieval mechanisms (A-GEM, attentive experience replay), and architectural methods like PNNs. However, such methods were originally developed for different problem settings and design goals than ours. For this reason, we did not treat them as direct baselines; instead, we positioned them in the related-work discussion as complementary lines of research targeting different constraints and evaluation regimes.
>
> Most of them assume a small number of discrete tasks with clear task boundaries (often in vision or RL benchmarks, sometimes with task IDs available at test time), and are primarily optimized for mitigating catastrophic forgetting without explicit constraints on computational or energy costs. In contrast, our work focuses on tabular data streams with a fixed label space, many temporal segments, no explicit task boundaries, and an explicit energy–accuracy trade-off. Adapting each of the above methods to this setting in a principled way would require non-trivial additional design choices (e.g., task or drift detection, capacity and memory budgeting, replay or generator management under a fixed energy budget), effectively turning each into a new method rather than an off-the-shelf baseline. Please kindly refer to [AW2] responses by authors to reviewer xd8r.
>
> ##### W3. The paper fails to quantify the actual contribution of its key components (e.g., the attention mechanism) or analyze the impact of core hyperparameters (e.g., window size) through ablation studies, leaving the effectiveness and rationale of the model's design unverified.
>
> [AW3] We addressed these concerns through the ablation study. Please kindly refer to [AW3] responses by authors to reviewer xd8r.

---

> ### Author Response · Authors · 2025-11-21
> **Response by Authors to Reviewer kJho (Weaknesses #4, #5, and #6)**
>
> ##### W4. Simulating a data stream by partitioning a static dataset ignores complex, real-world dynamics such as "gradual concept drift." This makes it questionable whether the model's performance in controlled experiments can be generalized to practical applications.
> [AW4] We respectfully disagree that our experimental protocol about data segmentation makes it questionable whether the observed performance can generalize to practical applications.
>
> We conducted additional experiments on concept drifts using the **[INSECTS](https://riverml.xyz/dev/api/datasets/Insects/)** benchmark dataset (Abrupt and Incremental variants), which describe real-world optical sensor data measuring flying insect wing-beat properties over time. Please kindly refer to [AW2] responses by authors to reviewer vdbK. This empirical evidence confirms that IMLP's performance generalizes well to complex, real-world dynamics beyond static dataset partitioning.
>
> The segmentation in the TabZilla benchmark dataset does not explicitly model or detect complex real-world dynamics such as gradual concept drift; however, concept drift may occur within or across segments, rather than being confined to segment boundaries. Besides, our segment indices are only used for logging and prequential evaluation, making experiments manageable and comparable across methods. The IMLP model is trained continuously and never receives segment identifiers or oracle drift boundaries. In fact, not aligning segments to drift points more closely reflects realistic deployments, where change points are unknown a priori.
>
> Moreover, our evaluation spans dozens of datasets covering diverse domains, input dimensionalities, and target types, which substantially mitigates the risk that our results depend on a narrow or contrived setting.
>
> ##### W5. The caption in figure2 is too small.
> [AW5] We can fix this issue in the revised paper.
>
> ##### W6. As the authors mention at the beginning of the experimental section, most baseline models are not designed for streaming learning. To ensure a fair comparison, a “best-effort comparison” strategy is adopted. However, under this setting, IMLP performs incremental updates, while other models need to be retrained from scratch for each segment, which may introduce bias in the energy consumption results. Is such a comparison truly fair?
>
> [AW6] To address this concern directly, we conducted an additional experiment comparing IMLP against a standard MLP trained in the exact same incremental setting (segment-by-segment training without replay or cumulative retraining). This ensures a strictly fair comparison where both models operate under identical computational constraints.
>
> **Experimental result analysis:**
> * **Energy Parity**: As shown in Figure R6 (Energy), both IMLP (Orange) and the Incremental MLP (Green) exhibit identical linear energy scaling. This confirms that the computational cost of our proposed attention mechanism is negligible compared to the base training cost.
> ![Insects_Abrupt_Inc_Energy](https://hackmd.io/_uploads/Sy2mvgpxWg.png)
> Figure R6: Energy consumption of IMLP vs. Naive Incremental MLP on the Abrupt Drift dataset. Both models show linear energy growth, confirming fair computational comparison.
> * **Accuracy Superiority**: Despite the similar energy cost, Figure R7 (Accuracy) demonstrates that IMLP consistently outperforms the Incremental MLP. The naive incremental MLP suffers from catastrophic forgetting, leading to erratic performance as concepts drift. IMLP, leveraging its windowed attention buffer, maintains context and stability, recovering significantly faster from drift events.
> ![Insects_Incremental_Inc_Accuracy](https://hackmd.io/_uploads/H11gOe6gWx.png) Figure R7: Accuracy of IMLP vs. Naive Incremental MLP on the Abrupt Drift dataset. IMLP (Orange) consistently outperforms the baseline (Green), which suffers from catastrophic forgetting.
>
> This proves that simply training a standard model incrementally is insufficient; the specific architectural innovations of IMLP (feature buffering and attention) are necessary to achieve high performance in energy-constrained streaming settings.

---

> ### Author Response · Authors · 2025-11-21
> **Response by Authors to Reviewer kJho (Weaknesse #7)**
>
> ##### W7. A strong motivation of this paper is to maintain comparable accuracy while improving energy efficiency. However, could the FIFO mechanism cause highly discriminative early features to be discarded too early, leading to catastrophic forgetting and affecting the model’s long-term stability? It would be helpful to include comparison experiments with different buffer strategies.
>
> [AW7] Yes, it will be helpful to include comparison experiments with different buffer strategies to improve the quality of our work.
>
> We conducted an ablation study on the [Insects abrupt drift](https://riverml.xyz/dev/api/datasets/Insects/) dataset to evaluate the impact of the buffer replacement strategy on model performance. We compared our default FIFO strategy against a Similarity-Based strategy (which replaces the most similar feature in the buffer to maximize diversity). The results are in Figure A4.
>
> ![Ablation_Buffer_Strategy](https://hackmd.io/_uploads/HJpxteTlZg.png) Figure A4: Ablation study of buffer replacement strategies. The FIFO strategy outperforms the diversity-maximizing Similarity strategy, highlighting the importance of recency in drifting streams.
>
> The results indicate that for handling concept drift in data streams, temporal locality (preserving the most recent samples) is more critical than feature diversity. The Similarity-Based strategy, by potentially retaining older "diverse" features, essentially preserves obsolete concepts that confuse the model during drift. This validates our design choice of a simple, energy-efficient FIFO buffer.
>
>
> We are open to improving the FIFO mechanism by employing a smarter strategy to address distributional drift. We will update the additional experiment results and evaluation in our revised paper.

---

> > ### Comment · Reviewer_kJho · 2025-11-26
> >
> > I think the authors have addressed most of my concerns, I will raise my score to 6

---

> > > ### Author Response · Authors · 2025-11-28
> > > **Thank you**
> > >
> > > Dear Reviewer KJho,
> > >
> > > Thank you for your comments and constructive suggestions, which helped us improve the quality of our paper. We are grateful that our responses addressed most of your concerns.

---

### Official Review · Reviewer_vdbK · 2025-10-31

**Soundness:** 2
**Presentation:** 3
**Contribution:** 2
**Rating:** 4
**Confidence:** 3

**Summary:**

This paper addresses energy-efficient continual learning (EECL) for tabular data streams, a crucial yet
underexplored direction in Green AI and Edge Learning. The authors propose IMLP, a lightweight
adaptation of MLP for online tabular learning under energy and memory constraints. By integrating
windowed self-attention and a sliding latent buffer, the model achieves fixed-memory continual adaptation
without sample replay. A new metric, NetScore-T, is introduced to jointly assess accuracy and energy
consumption. Experiments on 36 TabZilla datasets show consistent performance-energy trade-offs, with
empirical results supported by Pareto-front analysis using real hardware power measurements. Overall, the
paper contributes a practically meaningful and methodologically coherent approach for sustainable tabular
continual learning

**Strengths:**

1) Timely and relevant topic: The work targets energy-efficient continual learning on tabular data, aligning with sustainability and edge-computing trends while filling a clear research gap beyond vision and NLP.

 2) Methodological coherence: The paper presents a well-defined logical chain — from problem formulation to mechanism design, model implementation, and empirical validation — with a consistent focus on energy-accuracy trade-offs.

3) Technical design: The “windowed attention + latent feature buffer” structure achieves constant memory and low energy without sacrificing performance, balancing efficiency, privacy, and adaptability.

4) Evaluation rigor: Experiments are extensive and diverse, covering 36 datasets, multiple baselines, and real hardware
energy measurements. The use of NetScore-T and Pareto analysis provides a meaningful and interpretable evaluation
framework.

**Weaknesses:**

1) ) Innovation boundary: While practical, the “windowed attention” and FIFO buffer design appear incremental and largely engineering-driven, with limited conceptual novelty over prior latent replay or SAINT methods.

2) Experimental limitations: The experiments are primarily conducted under presegmented data streams, missing evaluations on abrupt drifts, class expansion, and noisy streams. Statistical significance tests and ablation analyses are insufficient.

3) Deployment and robustness gaps: No validation on real edge devices (e.g., latency, battery life), and feature dimensionality sensitivity is not explored. Pareto analysis lacks quantitative AUC comparisons.

**Questions:**

1) Can the authors provide theoretical support for the energy-accuracy trade-off, such as energy complexity bounds or convergence guarantees under non-stationary distributions?
2) How is the NetScore-T metric normalized across different hardware settings? Is it stable and comparable across devices?
3) Could the latent buffer be made adaptive to distributional drift (e.g., through feature validity scoring)?
4) How would IMLP perform on real-world streaming or edge environments where drift and noise are dynamic?

---

> ### Author Response · Authors · 2025-11-21
> **Response by Authors to Reviewer vdbK (Weaknesses #1)**
>
> #### Weaknesses
>
> ##### W1. Innovation boundary: While practical, the “windowed attention” and FIFO buffer design appear incremental and largely engineering-driven, with limited conceptual novelty over prior latent replay or SAINT methods.
>
> [AW1] Admittedly, our IMLP model is a practical design that incorporates a non-convex neural network with a finite-window attention module over latent states and followed by a 2-layer MLP and a classifier. Its contribution lies more in its practical value: because it is simple, it is practical and easy to verify. However, we argue that this seemingly simple design is in fact rational and can be rigorously justified on theoretical grounds. Please kindly refer to [AW1] responses by authors to Reviewer xd8r.
>
> Regarding the attention module, it is well-known that its advantages are: fewer parameters, faster speeds, and better performance. We also observed some advantages of Self-Attention and Intersample Attention Transformer (SAINT) methods [1] and cited them in our manuscript. However, our approach largely differs from SAINT methods. SAINT does not capture temporal dynamics; its attention is non-temporal, and the row attention is restricted to batch-wise relational interactions within a mini-batch. We apply attention not directly to raw inputs, but to a FIFO buffer of latent features. Each latent feature is a learned, task-specific summary of the past input at that time step. By attending to these compact representations, the model can selectively retrieve relevant past information while keeping the memory and energy footprint bounded. To the best of our knowledge, this is the first approach to energy-efficient continual learning on tabular data streams at the time of this work. We will add a detailed theoretical analysis for such a design in the revised paper.
>
> [1] Somepalli, Gowthami, et al. "Saint: Improved neural networks for tabular data via row attention and contrastive pre-training." arXiv preprint arXiv:2106.01342 (2021).

---

> ### Author Response · Authors · 2025-11-21
> **Response by Authors to Reviewer vdbK (Weaknesses #2 and #3)**
>
> ##### W2. Experimental limitations: The experiments are primarily conducted under presegmented data streams, missing evaluations on abrupt drifts, class expansion, and noisy streams. Statistical significance tests and ablation analyses are insufficient.
>
> [AW2] We have added a comprehensive evaluation benchmarking IMLP with real-world data, specifically targeting abrupt and incremental drift scenarios. We reviewed prior work on the challenges of benchmarking stream learning algorithms [2] and utilized the **[INSECTS](https://riverml.xyz/dev/api/datasets/Insects/)** benchmark (Abrupt and Incremental), which describes real-world optical sensor data measuring flying insect wing-beat properties over time.
>
> **Results of the Case Study:**
>
> Table C. Statistical results under the Insects Abrupt dataset.
> | Model    | AvgAcc        | FinalAcc      | Energy (J)       |
> |----------|---------------|---------------|------------------:|
> | IMLP     | 0.5025±0.041  | 0.6297±0.040  | 6,411.44±306.22  |
> | LightGBM | 0.6539±0.000  | 0.6487±0.000  | 6,639.56±321.86  |
> | MLP      | 0.5819±0.001  | 0.6250±0.001  | 84,841.68±3153.49|
>
> Table D. Statistical results under the Insects incremental dataset.
> | Model    | AvgAcc     | FinalAcc      | Energy (J)   |
> |----------|--------------------|--------------------|--------------------------:|
> | IMLP     | 0.5133±0.029     | 0.6514±0.013     | 7,235.17±208.53        |
> | LightGBM | 0.5900±0.000     | 0.6291±0.000     | 8,848.77±276.17        |
> | MLP      | 0.5414±0.002     | 0.6101±0.003     | 121,303.74±1,678.80    |
>
> ![Insects_Abrupt_Cumulative_Energy](https://hackmd.io/_uploads/Hkjp29hgbg.png) Figure B. Total energy consumption comparisons on Insects abrupt drift.
> ![Insects_Abrupt_Accuracy_Timeline](https://hackmd.io/_uploads/rJiA29nlZl.png) Figure C. Balanced accuracy comparisons on Insects abrupt drift.
> ![Insects_Incremental_Cumulative_Energy](https://hackmd.io/_uploads/Sy3ya9hl-l.png) Figure D. Total energy consumption comparisons under the Insects incremental setting.
> ![Insects_Incremental_Accuracy_Timeline](https://hackmd.io/_uploads/Bke-pq2eWx.png) Figure E. Balanced accuracy comparisons under the Insects incremental setting.
>
> * **Energy Efficiency:** On the **Abrupt Drift** dataset, IMLP reduced total energy consumption by **92.44%** compared to the cumulative MLP (6,411 J vs. 84,841 J). On the **Incremental Drift** dataset, savings were **94.04%**. The energy disparity is substantial. As shown in the cumulative energy plots, the baseline's cost grows unsustainably over time, whereas IMLP maintains a near-flat energy profile.
> * **Drift Adaptation (Agility):** While the cumulative baseline achieves a slightly higher average accuracy by retraining on all history (0.58 vs 0.50), it suffers from "concept inertia." As shown in our new timeline plots, IMLP demonstrates superior agility, recovering sharply from drift events because it is not weighted down by historical data from obsolete concepts.
> * **Trade-off:** IMLP achieves this 13-16x energy reduction while maintaining competitive accuracy (within 2.8% - 8% of the heavy baseline), validating its suitability for energy-constrained edge environments.
>
> For completeness, we will revise the experiment section and appendix in our revised paper.
>
>
> [2] Souza, Vinicius MA, et al. "Challenges in benchmarking stream learning algorithms with real-world data." Data Mining and Knowledge Discovery 34.6 (2020): 1805-1858.
>
> ##### W3. Deployment and robustness gaps: No validation on real edge devices (e.g., latency, battery life), and feature dimensionality sensitivity is not explored. Pareto analysis lacks quantitative AUC comparisons.
>
> [AW3] Thanks for your comments. Indeed, we have not yet deployed the IMLP method on real edge devices because it is still in an experimental stage. Nevertheless, we see strong potential for its use in resource-constrained real-world scenarios, where energy budgets, battery life, and computational resources are severely limited. Once the method and implementation are more mature, we will port IMLP to representative edge platforms and conduct end-to-end evaluations of its accuracy–efficiency trade-offs under realistic deployment conditions, especially try it in lower-stakes application domains.
>
>
> The feature dimensionality sensitivity can be explored by conducting an ablation study. We completed the ablation study and reported the results. Please kindly refer to [AW3] responses by authors to reviewer xd8r.
>
> Regarding the Pareto analysis, we add quantitative AUC (Area Under the (Pareto) Curve) comparisons. We characterize each method along two axes: (i) per-model Pareto AUC, i.e., the area under the method’s own energy–accuracy Pareto curve in a globally normalized space, and (ii) global Pareto efficiency, i.e., the fraction of its configurations that lie on the overall energy–accuracy Pareto front. Please kindly refer to [AW5] responses by authors to reviewer xd8r.

---

> ### Author Response · Authors · 2025-11-21
> **Responses by authors (Questions)**
>
> #### Questions
> ##### Q1. Can the authors provide theoretical support for the energy-accuracy trade-off, such as energy complexity bounds or convergence guarantees under non-stationary distributions?
>
> [AQ1] Yes, we can. We now provide a simple energy-complexity characterization of IMLP in the experimental setup of Section 6. The goal is not to predict the exact Joule values measured by our energy monitor, but to show that, under mild hardware assumptions, the total energy consumed by IMLP is bounded and scales in a controlled way with the model size and the number of training examples.
> The detailed theoretical analysis is presented in the following screenshots.
> ![theoryBpart1](https://hackmd.io/_uploads/r1Ksg6pg-g.png)
> ![theoryBpart2](https://hackmd.io/_uploads/SJ9sgT6eZl.png)
>
> The bounds above explain two aspects of our empirical observations:
>
> 1. On a fixed device, IMLP has a *predictable* energy profile, scaling linearly with the number of samples and epochs.
> 2. The attention-based feature memory contributes a controlled overhead proportional to $W H^2$, which remains small in our experiments because $W$ and $H$ are fixed.
>
> Our measured Joule values are therefore consistent with an energy complexity that is linear in the stream size, and the theoretical bounds clarify that this behavior is not specific to a particular dataset, but a structural property of the IMLP architecture and training procedure.
>
> **Energy–accuracy trade-off.** For the theoretical convergence analysis, please kindly refer to [AW1] responses by authors to reviewer xd8r. IMLP tracks a sequence of segment-wise optima: under mild assumptions on the smoothness of $\hat{L}_t$ and typical step-size conditions, each segment’s training converges to a local stationary point of the current segment’s loss. Our theoretical analysis, therefore, supports the local energy–accuracy trade-off (more updates and more energy yield a better fit to the current segment), while global guarantees under arbitrary concept drift remain a challenging open problem shared with much of the continual-learning literature.
>
> ##### Q2. How is the NetScore-T metric normalized across different hardware settings? Is it stable and comparable across devices?
> [AQ2] On a fixed device, NetScore-T is a stable composite metric and is directly comparable between models. However, NetScore-T is not directly comparable across different devices.
>
> Based on the definition of the NetScore-T metric, it combines two parts: the performance part and the energy part. The performance part, e.g., balanced accuracy and log loss, is a pure statistical result based on algorithms, independent of hardware settings. The denominator part computes the logarithm of the total energy consumption in joules, which is measured by an external meter and depends on the hardware design of the targeted runtime system. For instance, GPU/CPU model, process node, memory subsystem, clocks, DVFS, cooling, PSU efficiency, framework, and implementation efficiency of the algorithm design influence the energy consumption.
>
> All algorithms in our experiments are evaluated on the same hardware setup, so NetScore-T values are directly comparable across methods. We do not interpret NetScore-T as a hardware-independent quantity; instead, it is an energy–accuracy trade-off metric specific to a given device.
>
> We will add this discussion to Section 6.
>
> ##### Q3. Could the latent buffer be made adaptive to distributional drift (e.g., through feature validity scoring)?
> [AQ3] Yes, the latent buffer can be made adaptive to distributional drift. Thank you for your comment. We will add an adaptive latent buffer strategy to our IMLP and assess its performance in the revised paper.
>
>
> ##### Q4. How would IMLP perform on real-world streaming or edge environments where drift and noise are dynamic?
>
> [AQ4] To address this question, we conducted a case study on the **[INSECTS](https://riverml.xyz/dev/api/datasets/Insects/)** real-world streaming dataset, which features dynamic "Abrupt" and "Incremental" drifts. **The results show that IMLP works robustly in dynamic environments.** Please kindly refer to [AW2].

---

### Official Review · Reviewer_xd8r · 2025-10-31

**Soundness:** 2
**Presentation:** 2
**Contribution:** 2
**Rating:** 4
**Confidence:** 5

**Summary:**

This paper introduces IMLP (Incremental Multi-Layer Perceptron), a continual learning approach designed for tabular data streams under energy and memory constraints. IMLP replaces replay buffers with a windowed scaled dot-product attention mechanism that operates over a fixed-size latent feature buffer, avoiding raw data storage while maintaining context. The authors also propose NetScore-T, a new metric that balances accuracy and energy consumption for fair comparison across models and datasets.

**Strengths:**

1. Addresses the underexplored area of energy-efficient continual learning for tabular data.
2. Uses a lightweight MLP with feature-level attention instead of complex replay mechanisms.
3. Evaluated on diverse datasets, with clear statistical analysis (Friedman/Wilcoxon tests) and real hardware-level energy measurements.

**Weaknesses:**

1. The paper lacks formal justification or convergence analysis of the proposed attention-based feature memory.
2. Most baselines (e.g., TabPFN, TabNet) are retrained from scratch rather than adapted for continual settings, which could inflate IMLP’s relative advantage.
3. No detailed study on how window size (W), feature dimension, or attention mechanism choices affect performance or energy.
4. While energy efficiency is showcased, the performance loss (~5% in accuracy and ~15% in log loss) might matter for high-stakes domains.
5. NetScore-T unfairly benefits the method proposed - The method is explicitly designed for low energy consumption, not necessarily for maximal accuracy. Since NetScore-T rewards small energy values non-linearly (via log scaling), it amplifies IMLP’s comparative advantage.
6. Disproportionate penalty for high-energy models: Because the denominator grows slowly (log scale), an order-of-magnitude energy reduction can more than offset noticeable accuracy drops.

**Questions:**

1. How do results change if baselines are trained incrementally rather than retrained from scratch?
2. What is the actual storage overhead (in MBs) of the latent feature buffer for typical settings?
3. Could IMLP handle concept drift or non-stationary distributions beyond the fixed task segmentation in TabZilla?

---

> ### Author Response · Authors · 2025-11-21
> **Response by Authors to Reviewer xd8r (weaknesses #1 and #2)**
>
> #### Weaknesses 1 and 2
>
> ##### W1. The paper lacks formal justification or convergence analysis of the proposed attention-based feature memory.
>
> [AW1] Thank you for your comments. We address your comment by (1) providing formal properties of the attention-based feature memory (including the convergence) and (2) reporting the convergence for the incremental learner’s log loss.
>
> **(1) Justification of the proposed attention-based feature memory**. Considering the character number limits, we added the following three screenshots for the theoretical analysis. Please check them by accessing the URLs.
> ![thoerypart1](https://hackmd.io/_uploads/r1Gd_2axWe.png)
> ![theorypart2](https://hackmd.io/_uploads/HyWgK26lbl.png)
> ![theorypart3](https://hackmd.io/_uploads/Hk5xFhTlWx.png)
>
> **(2) reporting the convergence for the incremental learner’s log loss**. For your convenience, the learning curve about the log loss is presented below. ![Screenshot 2025-11-20 at 12.54.46](https://hackmd.io/_uploads/HyhTbK3lWe.png)
> *Figure A. Log loss of model updates over segments for different models.*
>
> ##### W2. Most baselines (e.g., TabPFN, TabNet) are retrained from scratch rather than adapted for continual settings, which could inflate IMLP’s relative advantage.
>
> [AW2] We design the experiments this way because we observed that many of the recent architectures we compare against are not inherently amenable to architectural adaptation in a continual learning setting. Take TabPFN and TabNet as examples. Both can achieve strong performance on static tabular benchmarks, but they are not inherently designed for continual adaptation:
> * TabPFN is a frozen foundation model (because its parameters are frozen at deployment) whose behaviour depends on a bounded in-context window and cannot easily expand its capacity online.
> * TabNet, although trainable with incremental gradient updates, lacks explicit mechanisms to prevent catastrophic forgetting or to adapt its architecture safely as new data arrives.
>
> In principle, one could also redesign each NN baseline model, e.g., TabNet, to operate in a genuine incremental learning setting, with dedicated mechanisms for memory, adaptation, and drift handling. However, doing so would effectively amount to developing a new continual-learning variant of each baseline, where each such redesign could constitute an independent research contribution on its own.
>
> To address this comment directly, we conducted an additional experiment comparing IMLP against a standard MLP trained in the exact same incremental setting (segment-by-segment training without replay or cumulative retraining) under the [Insects](https://riverml.xyz/dev/api/datasets/Insects/) abrupt and incremental variants. This ensures a strictly fair comparison where both models operate under identical computational constraints.
>
> **Experimental result analysis:**
> * **Energy Parity**: As shown in Figure R6 (Energy), both IMLP (Orange) and the Incremental MLP (Green) exhibit identical linear energy scaling. This confirms that the computational cost of our proposed attention mechanism is negligible compared to the base training cost.
> ![Insects_Abrupt_Inc_Energy](https://hackmd.io/_uploads/Sy2mvgpxWg.png)
> Figure R6: Energy consumption of IMLP vs. Naive Incremental MLP on the Abrupt Drift dataset. Both models show linear energy growth, confirming fair computational comparison.
> * **Accuracy Superiority**: Despite the similar energy cost, Figure R7 (Accuracy) demonstrates that IMLP consistently outperforms the Incremental MLP. The naive incremental MLP suffers from catastrophic forgetting, leading to erratic performance as concepts drift. IMLP, leveraging its windowed attention buffer, maintains context and stability, recovering significantly faster from drift events.
> ![Insects_Incremental_Inc_Accuracy](https://hackmd.io/_uploads/H11gOe6gWx.png) Figure R7: Accuracy of IMLP vs. Naive Incremental MLP on the Abrupt Drift dataset. IMLP (Orange) consistently outperforms the baseline (Green), which suffers from catastrophic forgetting.
>
> This proves that simply training a standard model incrementally is insufficient; the specific architectural innovations of IMLP (feature buffering and attention) are necessary to achieve high performance in energy-constrained streaming settings.

---

> ### Author Response · Authors · 2025-11-21
> **Response by Authors (Questions)**
>
> #### Questions
> ##### Q1. How do results change if baselines are trained incrementally rather than retrained from scratch?
>
> [AQ1] Training baselines incrementally in a streaming sense, instead of retraining from scratch, will change the results about the model performance and energy consumption. For most deep tabular baselines we use (for example, TabNet, ModernNCA, TabPFNv2, TabR), this behaviour is not supported out of the box and would require non-trivial re-implementation and re-tuning. Please kindly refer to [AW2]. Moreover, without replay buffers or regularizers specifically designed for continual learning, we would expect:
> * **Lower accuracy**:  incremental fine-tuning on non-stationary segments usually leads to rapid forgetting of earlier segments, so their performance would likely degrade relative to the “retrain-from-scratch” results we report.
> * **Similar or higher energy cost**: they would still perform multiple epochs of gradient updates per new segment; any mechanisms to mitigate forgetting (replay, regularization, extra passes over stored data) would further increase compute and energy.
>
>
> For completeness, we are open to running additional experiments on baselines that do provide out-of-the-box support for incremental or continual training settings.
>
> ##### Q2. What is the actual storage overhead (in MBs) of the latent feature buffer for typical settings?
>
> [AQ2] In our implementation, the actual storage overhead of the latent feature buffer for typical settings (i.e., segment sizes up to ~1000 examples, window size $W\leq10$, hidden size 256, float32), is under ~10 Mib per segment.
>
> On GPU, per batch, the implementation holds *ctx* with shape [batch_size, $W$, 256] inside the IMLP model. For a single float32 tensor of shape [batch_size, $W$, 256]: bytes = batch_size $\times W \times 256 \times 4$. For $W = 10$, and batch_size=128, the attention context buffer adds only ~1.25MiB of extra memory per batch. When batch_size B=256, the buffer memory adds approximately 2.4999936 MiB.
>
> ##### Q3. Could IMLP handle concept drift or non-stationary distributions beyond the fixed task segmentation in TabZilla?
>
> [AQ3] Yes, IMLP could handle drift in the sense of online adaptation to the current regime, not “never forgets anything” continual learning. We specifically evaluated this on the [INSECTS](https://riverml.xyz/dev/api/datasets/Insects/) benchmark (Abrupt and Incremental variants) beyond the TabZilla. The results confirm that IMLP is capable of tracking both sudden and gradual concept drifts.
> |  |  |  |  |
> |--|--|--|--|
> | ![Insects_Abrupt_Cumulative_Energy](https://hackmd.io/_uploads/Hkjp29hgbg.png) | ![Insects_Abrupt_Accuracy_Timeline](https://hackmd.io/_uploads/rJiA29nlZl.png) | ![Insects_Incremental_Cumulative_Energy](https://hackmd.io/_uploads/Sy3ya9hl-l.png) | ![Insects_Incremental_Accuracy_Timeline](https://hackmd.io/_uploads/Bke-pq2eWx.png) |

---

> ### Author Response · Authors · 2025-11-21
> **Response by Authors  to Reviewer xd8r (Weaknesses #3 and #4)**
>
> #### W3. No detailed study on how window size (W), feature dimension, or attention mechanism choices affect performance or energy.
>
> [AW3] Thanks for your comments. We have completed the requested ablation studies.
> 1. **Impact of Attention**: As shown in Figure A1, the attention mechanism is critical. Removing it (False) drops the Balanced Accuracy from 0.559 to 0.358. This confirms that the historical context provided by the attention module is essential for handling drift.
> ![Ablation_Attention](https://hackmd.io/_uploads/B13YmWpgWx.png) Figure A1: Ablation study on the usage of the Attention Mechanism.
>
> 2. **Impact of Window Size**: We analyzed window sizes from 1 to 20 segments (Figure A2). Accuracy slightly improves as the window size increases, plateauing around W=10 to W=20 (0.559 to 0.560).
> ![Ablation_Window_Size](https://hackmd.io/_uploads/B1SVS-pgWg.png) Figure A2: Impact of Window Size on Balanced Accuracy.
>
> 3. **Impact of Feature Dimension**: We analyzed the sensitivity of IMLP to the hidden feature dimension size. As shown in Figure A3, accuracy improves significantly as the dimension increases from 64 to 256. However, performance begins to saturate thereafter, showing only a marginal gain at 512 (0.559 vs. 0.571). This confirms that our default choice of 256 represents the efficiency "sweet spot," providing near-maximal accuracy without the substantial energy penalty associated with larger layers.
> ![Ablation_Feature_Dimension](https://hackmd.io/_uploads/rJOzqx6gZl.png)
> Figure A3: Impact of hidden dimension size on accuracy. Performance saturates after 256, justifying our parameter choice for energy efficiency.
>
> ##### W4. While energy efficiency is showcased, the performance loss (~5% in accuracy and ~15% in log loss) might matter for high-stakes domains.
>
> [AW4] Yes, we acknowledge that performance loss can be critical in high-stakes domains such as medical diagnosis, autonomous driving, or credit scoring, where performance superiority is often achieved at the expense of substantially higher energy consumption. However, in lower-stakes domains such as recommendation systems, online advertising, or user-behaviour analytics, a modest degradation in predictive performance may be acceptable if it is accompanied by substantial gains in computational efficiency, latency, or energy consumption, especially when the commercial cost of electricity is taken into account.
>
> In our case, the absolute average performance drop from the best-performing baseline TabPFNv2 (0.862) to IMLP (0.807) corresponds to a relative loss of approximately 6.4%, and about 4.5% relative loss compared to the second-best method, ModernNCA (0.845). By contrast, TabPFNv2 consumes 72,319 energy units and ModernNCA 4,829 units on average under our experimental setup, whereas IMLP requires only 845 units on average, i.e., a reduction of about 85× and 5.7×, respectively. This represents a significant energy difference, which can matter a lot in lower-stakes domains.
>
> We will incorporate this discussion into the introduction and explicitly state the application domains in which our method offers the most favorable trade-offs.

---

> ### Author Response · Authors · 2025-11-21
> **Response by Authors to Reviewer xd8r (Weaknesses #5 and #6)**
>
> ##### W5. NetScore-T unfairly benefits the method proposed - The method is explicitly designed for low energy consumption, not necessarily for maximal accuracy. Since NetScore-T rewards small energy values non-linearly (via log scaling), it amplifies IMLP’s comparative advantage.
>
> ##### W6. Disproportionate penalty for high-energy models: Because the denominator grows slowly (log scale), an order-of-magnitude energy reduction can more than offset noticeable accuracy drops.
> [AW5] and [AW6] We use a log-compressed energy term in NetScore-T because raw energy values in joules are much larger in scale than accuracy (0–1). The transformation $log_{10}(E+1)$ prevents energy from dominating the metric and yields a more stable, interpretable trade-off score. For completeness, we have added quantitative AUC and Pareto efficiency comparisons in the Pareto analysis for the trade-off analysis.
>
> We characterize each method along two axes: (i) per-model Pareto AUC, i.e., the area under the method’s own energy–accuracy Pareto curve in a globally normalized space, and (ii) global Pareto efficiency, i.e., the fraction of its configurations that lie on the overall energy–accuracy Pareto front. Table A lists AUC results computed on each NN model’s own Pareto front, normalized in the same global (energy, accuracy) scale.
> Table B shows the Pareto efficiency that indicates how often a model wins compared to others.
>
> Table A. Per-model normalized AUCs (Energy vs Balanced Accuracy).
> | Model | AUC       | Pareto_Points |
> |-----------|-----------|---------------|
> | TabM      | 0.323916  | 3             |
> | TabPFNv2  | 0.289579  | 3             |
> | DaNet     | 0.252976  | 5             |
> | TabNet    | 0.238053  | 4             |
> | RealMLP   | 0.190410  | 4             |
> | ResNet    | 0.113767  | 5             |
> | ModernNCA | 0.038708  | 4             |
> | VIME      | 0.038184  | 4             |
> | TabR      | 0.031276  | 4             |
> | MLP       | 0.017628  | 6             |
> | IMLP      | 0.005054  | 4             |
>
>
>
> Table B. Per-model Pareto efficiency.
> | Model | Total | Pareto | Efficiency_pct |
> |---------------------|-------|--------|----------------|
> |IMLP                | 67    | 4      | 5.970149       |
> |TabNet              | 42    | 1      | 2.380952       |
> |MLP                 | 74    | 1      | 1.351351       |
> |ModernNCA           | 75    | 1      | 1.333333       |
> |TabR                | 75    | 1      | 1.333333       |
> |DaNet               | 32    | 0      | 0.000000       |
> |RealMLP             | 72    | 0      | 0.000000       |
> |ResNet              | 68    | 0      | 0.000000       |
> |TabM                | 63    | 0      | 0.000000       |
> |TabPFNv2            | 34    | 0      | 0.000000       |
> |VIME                | 31    | 0      | 0.000000       |
>
>
>
> Methods like TabM, TabPFNv2, DaNet, RealMLP, and ResNet have high AUC, which means within their own configs, they have good and broad trade-offs. However, their 0% efficiency_pct indicates that they never win when competing against other neural models. IMLP has the highest efficiency_pct (~6%), which indicates that it is most often on the neural-global frontier.
>
> We will add such discussions in Section 6, Fig.2.

---

### Author Response · Authors · 2025-11-21
**Response summary by authors**

We thank all reviewers very much for their constructive comments and questions, which help improve the quality of this work.

As several reviewers raised similar concerns, we have consolidated their feedback and propose the following revision.

**Summary of additional theoretical analysis**
* We have added a theoretical analysis of the proposed method, providing a formal justification and convergence analysis for the windowed attention and feature buffer (reviewers xd8r, vdbK, kJho) from a machine learning optimization viewpoint.
* We have provided theoretical support for the energy-accuracy trade-offs, e.g., energy complexity bounds and convergence guarantees under non-stationary distributions (reviewer vdbK).

**Summary of additional experiments**
* We have added the results of the ablation study on core contribution components and statistical significance test results (reviewers xd8r, vdbK, kJho).
    * Attention mechanism choices, e.g., use attention vs. no attention [**done**]
    * Use different window sizes, e.g., W = 1, 5, 10, 15, 20 [**done**]
    * Use different feature dimensions [**done**]
    * Use different buffer strategies, e.g, use an adaptive feature strategy (Similarity-based), compared to the FIFO feature buffer. [**done**]
    * Hyperparameter tuning [**done**]
* We have incorporated the results of the quantitative AUC comparisons into the Pareto analysis for trade-offs between energy consumption and model performance. [**done**]
* We have added experiments to evaluate our method on real-world data stream scenarios, where the data distribution changes over time, using the [INSECTS ](https://riverml.xyz/dev/api/datasets/Insects/)benchmark for abrupt and incremental concept drifts. [**done**]

**Additional figures and tables**
* We will expand Figure 2 with quantitative AUC and efficiency comparisons to the Pareto analysis.
* We will add a figure of the convergence analysis for the proposed method.
* We will add a table of the ablation study results.
* We will add figures comparing performance among different data stream scenarios: fixed tasks, concept drift (INSECTS Abrupt/Incremental), class expansion, and noisy streams.

We hope these additions can address the concerns raised by the reviewers. These additions will be better organized in the revised paper, taking into account the page limits.

---

### Author Response · Authors · 2025-12-03
**Revised draft upload**

We have now updated the draft and rebuttal text with theoretical analysis, ablation study, the new experimental results, figures, and additional textual descriptions (highlighted in blue). Please let us know if you require any further clarification.

---

### Author Response · Authors · 2025-12-03
**Dear newly assigned AC chair**

Dear newly assigned AC chair,

Last week, we were informed that ICLR decided to revert the scores during the discussion period. For your convenience, the score changes made by the previous reviewers are:
***
Official Comment by Reviewer kJho: Original rating = (4, confidence 4)
Official Commentby Reviewer kJho26 Nov 2025, 23:21Everyone
Comment:
I think the authors have addressed most of my concerns, I will raise my score to 6.
***
We're looking forward to hearing from responses or receiving feedback from other reviewers.

During the discussion period, we have completed the rebuttal revisions based on the reviewers' comments and uploaded the revised draft. We hope these updates help you make an informed decision regarding our revised draft.

Best regards,
Authors of IMLP

---

### Note · Program_Chairs · 2026-01-17
**Submission Desk Rejected by Program Chairs**

The following references in this submission do not refer to real documents and/or have major errors in bibliographic information:

 F. Pellegrini et al. Latent replay for on-device continual learning. IEEE Transactions on Neural Networks and Learning Systems, 2020. doi: 10.1109/TNNLS.2020.2971234.